# A Large Group Emergency Decision-Making Method Based on Uncertain Linguistic Cloud Similarity Method

Gang Chen [1], Lihua Wei [2], Jiangyue Fu [1], Chengjiang Li [1,3] and Gang Zhao [3,*]

1. School of Management, Guizhou University, Guiyang 550025, China
2. School of Business, Sun Yat-sen University, Guangzhou 510275, China
3. School of Engineering, University of Tasmania, Hobart, TAS 7005, Australia
* Correspondence: gang.zhao@utas.edu.au

**Abstract:** In recent years, the consensus-reaching process of large group decision making has attracted much attention in the research society, especially in emergency environment area. However, the decision information is always limited and inaccurate. The trust relationship among decision makers has been proven to exert important impacts on group consensus. In this study, we proposed a novel uncertain linguistic cloud similarity method based on trust update and the opinion interaction mechanism. Firstly, we transformed the linguistic preferences into clouds and used cloud similarity to divide large-scale decision makers into several groups. Secondly, an improved PageRank algorithm based on the trust relationship was developed to calculate the weights of decision makers. A combined weighting method considering the similarity and group size was also presented to calculate the weights of groups. Thirdly, a trust updating mechanism based on cloud similarity, consensus level, and cooperation willingness was developed to speed up the consensus-reaching process, and an opinion interaction mechanism was constructed to measure the consensus level of decision makers. Finally, a numerical experiment effectively illustrated the feasibility of the proposed method. The proposed method was proven to maximally retain the randomness and fuzziness of the decision information during a consensus-reaching process with fast convergent speed and good practicality.

**Keywords:** uncertain linguistic information; clouds model; trust update; consensus level; clustering algorithm

## 1. Introduction

A series of emergencies broke out over the past few decades. A multitude of examples may be listed, for instance, the appearance of SARS in 2003, the 7.8-magnitude earthquake of Wenchuan in 2008, the explosion of Tianjin Port in 2015, and the spread of COVID-19 in 2019. The consequences are dire and long lasting. These disasters not only took a heavy toll on life and work but also were linked to substantial casualties and significant property losses. Generally, a decision must be made in a very short time if emergencies break out, which means the decision information is extremely limited and inaccurate [1–3]. Therefore, traditional decision-making methods cannot properly solve emergency decision making (EDM) problems, which has emerged as a heated topic of discussion and has recently gained increasing attention from both scholars and researchers [4–6].

Faced with limited, uncertain information as well as the inherent ambiguity of human thinking, many researchers have made attempts to cope with EDM problems by uncertain linguistic methods, including fuzzy sets [7,8] and linguistic terms [9–12]. Uncertain linguistic methods are a desirable choice because decision makers can qualitatively express information [13–15]. A wide range of studies have been devoted to using ambiguity in uncertain language [16,17]; however, researchers ignored the randomness of language evaluation. In an attempt to consider both randomness and fuzziness of language evaluation, some scholars have applied a cloud model that can effectively combine them in an

aggregation process to address decision problems [18–25]. On the basis of their significant contributions, we used cloud similarity to divide large-scale decision makers into several groups; moreover, the expected relative distance of the cloud was presented to measure the consensus level of decision makers.

Multi-target, multi-dimensional attributes, together with randomness, are common distinguishing features of EDM problems [26,27]. Moreover, as it has become increasingly likely to participate in the decision-making process for experts with the development of network technology, group joining in decision making has turned into a large group with multi-dimensional attributes [28,29]. The traditional group decision-making (GDM) method that focuses on a small number of experts is insufficient for major emergencies [30]. Large group emergency decision making (LGEDM) boasts a distinct advantage in that decision makers from various sectors and professional fields can solve EDM problems from different perspectives, which can ensure the globality and systematic nature of the results [31,32]. Nevertheless, it also has an obvious disadvantage in that the greater the difference in experts' knowledge structure, making the consensus level on the opinions among experts even lower, the more difficult it is for them to reach a consensus on opinions at once.

Making a decision satisfied through the consensus-reaching process (CRP) plays an indispensable role in addressing LGEDM problems [33–35], bringing studies for consensus-reaching process to the forefront in the field of large group decision making (LGDM). To this end, a number of researchers have constructed moderately reasonable consensus models from their own perspectives, providing strong support for consensus research [36–45].

As social networks develop by leaps and bounds, the way in which to use social relationships to improve consensus has become a new cutting-edge topic in the research field of LGDM. There are quite a few directions for social network analysis (SNA), such as conflict relationship [46,47], feedback behavior [48,49], non-cooperative behavior [40–54], cooperative behavior [55], and trust relationship [56–58]. Trust relationship plays an important role in social networks. Previously, trust relationship has been widely used in GDM to infer incomplete individual opinions, influence decision-making interactions, and improve the level of group consensus. Recently, some scholars have introduced trust relationship into CRP to group decision makers. However, few studies have developed a consensus model from the perspective of trust update. Under the circumstance of EDM, the trust relationship among decision makers is prone to being affected by consensus level, similarity of opinions, and willingness to modify opinions. Updating trust can reflect the trust relationship more accurately at different stages; it also can effectively adjust the weights of decision makers to reach consensus at a faster pace. Thus, we could make responses promptly to emergencies and reduce the decision cost. Therefore, this study proposes a novel consensus model with the trust-updating mechanism.

The rest of this study is structured as follows: A systematic literature review is found in Section 2. Section 3 presents some preliminaries about uncertain linguistic and cloud models. Section 4 introduces the consensus-reaching method and trust-updating approach for large group emergency decision making with uncertain linguistic information. Section 5 provides a hypothetical application and some simulation analyses to justify the proposed method. The final section presents conclusions and directions for future study.

## 2. Literature Review

To further elucidate the motivation of our proposal, related studies are reviewed in this section. For better illustration and explanation, we conducted our literature review from three aspects, i.e., decision information types, CRPs in GDM, and SNAs in CRP.

### 2.1. Decision Information Types

In order to deal with the uncertainty of the decision-making environment, researchers have proposed several decision information types, including fuzzy sets and linguistic terms. The types of fuzzy sets mainly include general fuzzy sets [7,8,34], fuzzy soft sets [3], asym-

metric fuzzy [14], intuitionistic fuzzy [36], and hesitant fuzzy [38], among others. Despite fuzzy sets being able to describe the uncertainty of decision information, most decision makers still prefer to utilize linguistic terms in evaluation. The types of linguistic terms mainly include general uncertain linguistic [9,13,16,31], 2-tuple linguistic [4,6], unbalanced linguistic [21], probabilistic linguistic [27], and hesitant fuzzy linguistic [10,12,17,41,43], among others. Decision makers could qualitatively express information by linguistic terms, but the randomness of language evaluation was ignored.

The cloud model is an uncertainty transformation model between a qualitative concept and its quantitative expression, providing attention to both fuzziness and randomness of language evaluation. Some researchers have applied the cloud model to the transformation of linguistic terms in decision-making problems. For instance, Wang et al. [9] presented a new similarity measure between clouds and a consensus-based method based on the cloud model for LGDM. Wang et al. [18] proposed several new aggregation operators that were based on the cloud model with linguistic information. Wang et al. [21] developed an asymmetric-trapezoidal-cloud-based linguistic GDM method under unbalanced linguistic distribution assessments. Compared with traditional methods of dealing with fuzzy concepts, cloud models are more intuitive and specific. However, there have only been a few studies on the application of the cloud model to the transformation of linguistic terms thus far.

### 2.2. CRPs in GDM

CRP plays an important role in GDM, aiming to help multiple decision makers adjust their opinions and obtain a result that can be accepted by most of them. Generally, a CRP includes two phases, i.e., consensus measurement and feedback adjustment. Researchers have proposed a variety of consensus measurement methods and feedback adjustment strategies from different perspectives. For example, Xu et al. [2] considered non-cooperative behaviors and minority opinions in CRP of LGEDM. Gou et al. [10] developed a consensus reaching process for LGDM with double hierarchy hesitant fuzzy linguistic preference relations. Rodriguez et al. [17] presented a new cohesion-driven CRP approach to deal with comparative linguistic expressions. Wang et al. [31] developed an objective adjustment coefficient on the basis of the expert's importance degree to modify the individual preferences in CRP. Zhong et al. [33] proposed a multi-stage hybrid feedback mechanism to improve the CRP, considering both cardinal consensus and ordinal consensus. Tang et al. [39] proposed an ordinal consensus measure with an objective threshold based on preference orderings. Long et al. [40] developed two-stage consensus-reaching models with minimum adjustments based on preference–approval structure and prospect theory. Li et al. [41] proposed a consensus model with a hierarchical feedback mechanism for LGDM dealing with hesitant fuzzy linguistic information. Tang et al. [42] proposed a hierarchical consensus model considering non-cooperative behaviors. Ren et al. [43] considered the individual acceptable consistency, combined with the group consensus of hesitant fuzzy linguistic preference relations in CRP. Wan et al. [44] developed a personalized individual semantic-based CRP. Wan et al. [45] proposed the concept of the lowest consensus threshold and presented a two-stage CRP method.

To summarize, the methods of CRP can be roughly divided into three categories: cooperative or non-cooperative behavior, preference relations, and feedback mechanisms. Recently, LGDM in the social network context has become an attractive research hotspot in decision science. Some researchers have applied SNA to CRP, and details are presented in Section 2.3.

### 2.3. SNAs in CRP

The relationship among decision makers in a social network is an important factor that can affect the results of decisions. The decision makers are homogeneous and can be regarded as nodes in a social network. Therefore, SNA methods can be used to determine the centrality or importance of each decision maker in CRP. For instance, Zhang et al. [35]

developed a feedback mechanism in CRP considering the leadership and the bounded confidence levels of experts. Ding at al. [46] presented an SNA-based conflict relationship investigation process and proposed a CRP based on conflict degree. Liang et al. [49] used the network DeGroot model to adjust the preferences of decision makers in the feedback mechanism of CRP. Bai at al. [55] proposed a CRP model with cooperative behavior that was based on social network analysis considering the propagation of decision-makers' preference. Zhao et al. [56] proposed a CRP method based on integrated relationships between trust and the similarity of opinions. Zhou et al. [57] took account of distributed linguistic trust relations under SNA to design a CRP. Peng et al. [58] developed a consensus detection model that was based on the incomplete trust relationship of social networks. It can be seen that trust relationship is the most frequently considered element by researchers in SNA.

Table 1 shows the comparison of the related literature. It can be seen that LGDM was the most popular decision-making type. Real numbers and hesitant fuzzy linguistic were the two most common types of decision information. When the decision information type was linguistic terms, a limited amount of the studies in the literature considered using the cloud model to transform language evaluation into its quantitative expression. Most of the studies in the literature researched the CRP in GDM or LGDM problems from different perspectives, but only a few studies utilized SNA methods to explore CRP.

**Table 1.** The comparison of the related literature.

| References | Decision-Making Types | Decision Information Types | CRP | SNA |
|---|---|---|---|---|
| Xu et al. [2] | LGEDM | Real numbers | Non-cooperative behaviors and minority opinions | × |
| Wang et al. [9] | LGDM | Linguistic terms + cloud model | Cloud similarity | × |
| Gou et al. [10] | LGDM | Hesitant fuzzy linguistic | Double hierarchy hesitant fuzzy linguistic preference relations | × |
| Rodriguez et al. [17] | LGDM | Hesitant fuzzy linguistic | Cohesion among the sub-group members | × |
| Wang et al. [18] | GDM | Linguistic terms + cloud model | × | × |
| Wang et al. [21] | GDM | Unbalanced linguistic + cloud model | × | × |
| Wang et al. [31] | LGDM | Linguistic terms | Objecive adjustment coefficient | × |
| Zhong et al. [33] | LGDM | Real numbers | Multi-stage hybrid feedback mechanism | × |
| Zhang et al. [35] | GDM | Interval fuzzy preference relations | Leadership and the bounded confidence | Network partition algorithm |
| Tang et al. [39] | LGDM | Heterogeneous preference information | Objective threshold | × |
| Long et al. [40] | GDM | Real numbers | Preference–approval structures in prospect theory | × |
| Li et al. [41] | LGDM | Hesitant fuzzy linguistic | Hierarchical feedback mechanism | × |
| Tang et al. [42] | GDM | Real numbers | Non-cooperative behaviors and minimum spanning tree | × |
| Ren et al. [43] | GDM | Hesitant fuzzy linguistic | Individual acceptable consistency and linguistic preference relations | × |

**Table 1.** *Cont.*

| References | Decision-Making Types | Decision Information Types | CRP | SNA |
|---|---|---|---|---|
| Wan et al. [44] | LGDM | Probabilistic linguistic preference relations | Personalized individual semantic | × |
| Wan et al. [45] | GDM | Linguistic intuitionistic fuzzy variables | Lowest consensus threshold | × |
| Ding et al. [46] | LGDM | Intuitionistic fuzzy values | Conflict degree | Conflict relationship investigation process |
| Liang et al. [49] | LGDM | Real numbers | Overconfident or unconfident behaviors | Social network DeGroot model |
| Bai et al. [55] | LGDM | Real numbers | Cooperative behaviors | Propagation of decision makers' preference |
| Zhao et al. [56] | LGDM | Real numbers | Integrated relationship network | Trust–opinion similarity relationships |
| Zhou et al. [57] | GDM | Linguistic terms | Complete interval distributed preference relation | Distributed linguistic trust relations |
| Peng et al. [58] | LGDM | Picture fuzzy numbers | Picture fuzzy Jensen a-norm dissimilarity measure | Incomplete trust relationship |
| This paper | LGEDM | Linguistic terms + cloud model | Opinion interaction and trust updating mechanisms | Trust relationship |

As discussed above, LGEDM methods still suffer several limitations despite the extensive research on this topic. Therefore, our study sought to explore the following motivations: (1) Most of the existing research focused on ordinary GDM or LGDM, yet LGDM in an emergency environment has barely been discussed. (2) Most of the existing LGDM methods assume that decision makers can provide accurate evaluation information, which is typically not the case in the emergency decision-making environment. Some of the existing LGDM methods assume that decision makers can provide evaluation information by hesitant fuzzy linguistic, which is reasonable and operable. However, the randomness of language evaluation is ignored. (3) Most of the existing LGDM methods only take account of decision makers' behaviors or preference relations in CRP, ignoring the relationship among decision makers in a social network, which could affect the results of decisions.

On the basis of the motivations mentioned above, the purpose of this study was to propose an uncertain linguistic cloud similarity LGEDM approach in a trust-relationship-based social network environment. The contributions of this study are summarized below:

(1) An improved method of converting linguistic information into cloud models was proposed to adapt to the continuous language environment, in light of the method presented by existing research only being suitable for discrete language values.

(2) A trust updating mechanism was developed to speed up the CRP, which took account of cloud similarity and consensus level, along with cooperation willingness.

(3) An opinion interaction mechanism considering consensus level and trust degree was constructed, introducing the expected relative distance of the cloud to measure the consensus level of decision makers.

(4) A novel uncertain linguistic cloud similarity LGEDM approach in a trust-relationship-based social network environment was proposed to solve emergency decision-making problems. Furthermore, a case study was provided under the emergency environment to demonstrate the effectiveness and feasibility of the proposed approach. Additionally, sensitivity and comparison analyses were conducted to verify the superiority of the proposed approach.

### 3. Preliminaries

*3.1. Linguistic Representation Modeling*

In the EDM environment, it is difficult for decision makers to quantitatively express their preferences for attributes, so they often qualitatively express their preferences with linguistic variables. Plenty of studies produced a mapping relationship to convert linguistic terms into numerical values to represent the semantics of linguistic terms. The following contents illustrate linguistic representation models.

**Definition 1** ([59]). Let $S = \{s_\theta | \theta = 1, 2 \cdots , g - 1, g - 1 \in N^+\}$ denote a linguistic term set (LTS) with odd cardinality, where $s_\theta$ is a possible linguistic term, and $g - 1$ is an odd number and generally takes 3, 5, 7, 9 and so on. Any $s_a, s_b \in S$ satisfies the following conditions:

(1) If $a > b$, then $s_a > s_b$;
(2) If the negative operator $neg(s_a) = s_b$, then $b = g - 1 - a$;
(3) If $s_a \geq s_b$, then $\max(s_a, s_b) = s_a$; if $s_a \leq s_b$, then $\min(s_a, s_b) = s_a$.

**Definition 2** ([60,61]). Let $\ddot{s} = [s_a, s_b]$, $s_a, s_b \in S$, and $s_a \leq s_b$; $s_a, s_b$ are the lower and upper limits of $\ddot{s}$, respectively. Then, we call $\ddot{s}$ uncertain linguistic values.

Assuming that $\ddot{s}_1 = [s_{a_1}, s_{b_1}]$ and $\ddot{s}_2 = [s_{a_2}, s_{b_2}]$ are uncertain linguistic values, $\vartheta \in [0, 1]$, then the following algorithm holds:

(1) $\ddot{s}_1 \oplus \ddot{s}_2 = [s_{a_1}, s_{b_1}] \oplus [s_{a_2}, s_{b_2}] = [s_{a_1+a_2}, s_{b_1+b_2}]$;
(2) $\vartheta \ddot{s}_1 = \vartheta [s_{a_1}, s_{b_1}] = [\vartheta s_{a_1}, \vartheta s_{b_1}] = [s_{\vartheta a_1}, s_{\vartheta b_1}]$.

*3.2. Cloud Models*

Some studies have noticed that a number of scholars tend to ignore the randomness of uncertain language in the process of using them. To reduce the information loss and distortion in the conversion process, some researchers consider it necessary to convert uncertain linguistic information into cloud models illustrated below.

**Definition 3** ([62]). Let $U$ be the universe of discourse and $C$ be a qualitative concept in $U$. If the membership degree $\mu_C(x)$ of the element $x \in U$ to the qualitative concept $C$ is a random number, then the distribution of membership degree $\mu_C(x)$ in the universe $U$ of discourse is called the membership cloud, hereinafter referred to as the cloud, namely, $\mu_C(x) \to [0, 1]$; each $x$ $(x \in U)$ has $x \to \mu_C(x)$.

When the distribution of the membership degree $\mu_C(x)$ in the universe $U$ of discourse features normal distribution, it is called a normal cloud. The normal cloud model can be denoted as $C(Ex, En, He)$, where the expectation $Ex$ represents the central value of the cloud droplet in the universe $U$ of discourse, entropy $En$ refers to the degree of ambiguity of the qualitative concept, and hyper-entropy $He$ refers to the uncertainty of entropy.

**Definition 4** ([63]). Let $C_1(Ex_1, En_1, He_1)$ and $C_2(Ex_2, En_2, He_2)$ be two adjacent one-dimensional normal clouds, and $Ex_1 \leq Ex_2$; then, the numerical characteristics of the comprehensive cloud $C(Ex, En, He)$ are calculated as follows:

(1) If $|Ex_2 - Ex_1| \geq 3|En_2 - En_1|$, there is
$Ex = [(Ex_1 + 3En_1) + (Ex_2 - 3En_2)]/2$,
$En = \max\{(Ex - Ex_1)/3, (Ex_2 - Ex)/3\}$,
$He = \sqrt{He_1^2 + He_2^2}$.
(2) If $|Ex_2 - Ex_1| < 3|En_2 - En_1|$, there is
$Ex = (Ex_1 En_2 + Ex_2 En_1)/(En_1 + En_2)$,
$En = \max\{En_1 + (Ex - Ex_1)/3, En_2 + (Ex_2 - Ex)/3\}$,
$He = \sqrt{He_1^2 + He_2^2}$.

**Definition 5** ([18]). Let $C_i(i = 1, 2, \cdots, n)$ be a group of clouds, and its corresponding weight vectors $w = (w_1, w_2, \cdots, w_n)$ satisfy $w_i \in [0, 1]$ $(i = 1, 2, \cdots, n)$ and $\sum_{i=1}^{n} w_i = 1$; then, the result obtained by aggregation *CWAA* is still a cloud:

$$CWAA_w(C_1, C_2, \cdots, C_n) = (\sum_{i=1}^{n} w_i Ex_i, \sqrt{\sum_{i=1}^{n} w_i En_i^2}, \sqrt{\sum_{i=1}^{n} w_i He_i^2}) \tag{1}$$

**4. Large Group Decision-Making Mechanism with Trust Update and Consensus Level**

*4.1. Framework*

Let us consider the emergency decision-making problem of a large group consensus with uncertain language information, in which

(1) There is a set of evaluation rounds, denoted by $T = \{1, 2, \cdots, h\}$, where $t$ denotes the $t$th round, $t = 1, 2, \cdots, h$.

(2) There is a set of decision makers, denoted by $E = \{e_1, e_2, \cdots, e_m\}$, where $e_i$ denotes the $i$th expert, $i = 1, 2, \cdots, m$. Moreover, the weight of decision-maker $i$ in round $t$ is $w_i^t$ and $\sum_{i=1}^{m} w_i^t = 1, w_i^t \geq 0$.

(3) There is a set of alternatives, denoted by $X = \{x_1, x_2, \cdots, x_n\}$, where $x_j$ denotes the $j$th alternative, $j = 1, 2, \cdots, n$.

(4) Decision makers use the uncertain linguistic for expressing their preferences of alternatives. The uncertain linguistics provided by all decision makers are denoted by $A^t = [s_{ij}^t]_{m \times n}$, where $s_{ij}^t = [s_{\theta_{ij}^t}^L, s_{\theta_{ij}^t}^U]$ is evaluation information of expert $i$ on alternative $j$ in round $t$. $s_{\theta_{ij}^t}^L, s_{\theta_{ij}^t}^U$ represent the upper and lower limits of the uncertain language value, respectively. $S = \{s_0, s_1, \cdots, s_g\}$ is a pre-defined granular language evaluation set, and $s_{\theta_{ij}^t}^L, s_{\theta_{ij}^t}^U \in S$.

(5) Experts use real numbers from 0 to 1 to express social trust relationships. The trust relationship across a group of all decision makers is denoted by $B^t = [b_{li}^t]_{m \times m}$, where $b_{li}^t$ denotes the trust value of expert $l$ to expert $i$ in the $t$th round.

Herein, we consider how to take full advantage of randomness and fuzziness of uncertain language and obtain an acceptable consensus level among experts by considering trust update. The resolution framework of the proposed large group consensus decision-making problem is shown in Figure 1, which mainly includes four stages, namely, convert cloud model, cluster decision makers, determining of the weight of decision makers, and the consensus-reaching process.

*4.2. Transformation of the Cloud Model*

Uncertain language should be transformed into a cloud model in order to effectively utilize the randomness and fuzziness of uncertain language. The language evaluation scale generally uses an odd number ($g$ is 2, 4, 6, 8, and so on) for group decision-making problems in a language environment with uncertain granularity. This paper presents an improved approach to generate a normal cloud with granularity language because the method presented by Wang et al. [9] is only suitable for discrete language values, not for continuous ones, if decision-makers define the universe of discourse as $[X_{\min}, X_{\max}]$. In this study, Formulas (2)–(5) illustrate the method how the uncertain language evaluation matrix $A^t = [s_{ij}^t]_{m \times n}$ transforms into normal cloud matrices $R_{ij}^{tL} = [C_{ij}^{tL}(Ex_{ij}^{tL}, En_{ij}^{tL}, He_{ij}^{tL})]_{m \times n}$ and $R_{ij}^{tU} = [C_{ij}^{tU}(Ex_{ij}^{tU}, En_{ij}^{tU}, He_{ij}^{tU})]_{m \times n}$. Then, we use Definition 4 to convert the cloud matrices $R_{ij}^{tL}$ and $R_{ij}^{tU}$ to a comprehensive cloud $R_{ij}^t = [C_{ij}^t(Ex_{ij}^t, En_{ij}^t, He_{ij}^t)]_{m \times n}$.

$$\varsigma(\theta_{ij}^t) = \begin{cases} \dfrac{\vartheta^{\frac{g}{2}} - \vartheta^{\frac{g}{2}-\theta_{ij}^t}}{2\vartheta^{\frac{g}{2}}-2}, & if\ 0 \le \theta_{ij}^t \le \frac{g}{2} \\ \dfrac{\vartheta^{\frac{g}{2}} + \vartheta^{\theta_{ij}^t - \frac{g}{2}}-2}{2\vartheta^{\frac{g}{2}}-2}, & if\ \frac{g}{2} < \theta_{ij}^t \le g \end{cases} \tag{2}$$

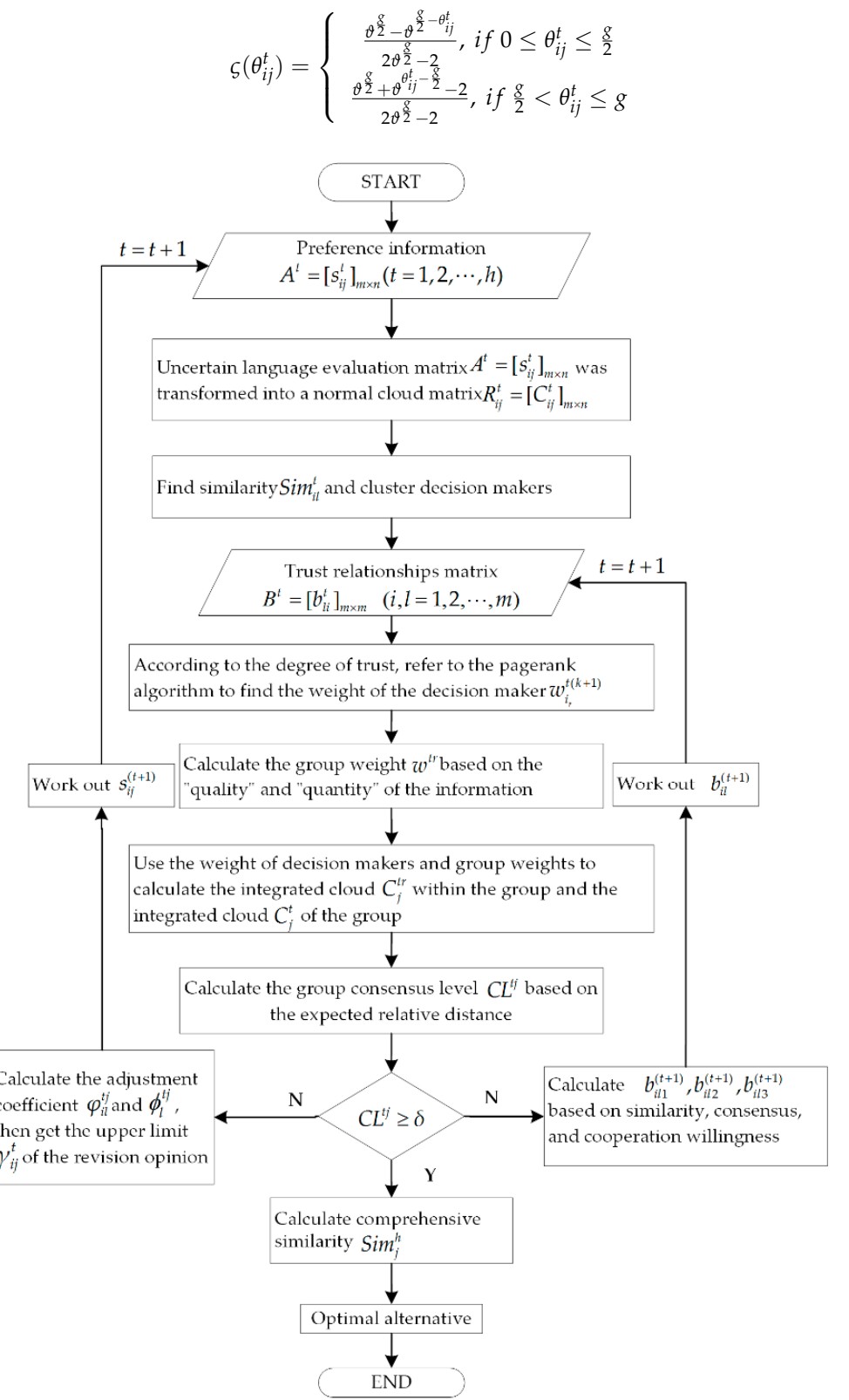

**Figure 1.** Large group decision-making process considering trust update.

It is worth noting that $\vartheta$ is equal to 1.37 according to Bao et al. [64] and Wang et al. [9].

$$Ex(\theta_{ij}^t) = X_{\min} + \varsigma(\theta_{ij}^t)(X_{\max} - X_{\min}) \tag{3}$$

$$En(\theta_{ij}^t) = \begin{cases} \dfrac{\varsigma(\theta_{ij}^t)(X_{\max}-X_{\min})}{6}, & if\ \theta_{ij}^t = \frac{g}{2} \\ \dfrac{En(g/2)}{1-(0.5-\varsigma(\theta_{ij}^t))^{2/g}}, & if\ 0 < \theta_{ij}^t < \frac{g}{2} \\ \dfrac{En(g/2)}{1-(\varsigma(\theta_{ij}^t)-0.5)^{2/g}}, & if\ \frac{g}{2} < \theta_{ij}^t < g \end{cases} \tag{4}$$

$$He(\theta_{ij}^t) = \begin{cases} \dfrac{He(g/2)}{1-(0.5-\varsigma(\theta_{ij}^t))^{2/g}}, & if\ 0 < \theta_{ij}^t < \frac{g}{2} \\ \dfrac{He(g/2)}{1-(\varsigma(\theta_{ij}^t)-0.5)^{2/g}}, & if\ \frac{g}{2} < \theta_{ij}^t < g \end{cases} \tag{5}$$

### 4.3. Cluster Grouping of Large Group Members

Large group clustering is grounded in the similarity of decision-making information among members. The clustering process should ensure the greatest similarity of opinions within the group, while the greatest difference among groups. We used the cloud similarity of group opinions to cluster large groups and a cloud similarity measure proposed by Wang et al. [9] to calculate the opinion similarity amongst decision makers.

$$Sim(C_{ij}^t, C_{lj}^t) = 1 - \frac{\left|\hat{s}(d(C_{ij}^t, C_{lj}^t))\right|}{\hat{s}(C_{ij}^t) + \hat{s}(C_{lj}^t)} \tag{6}$$

$$Sim_{il}^t = \frac{1}{n}\sum_{j=1}^{n} Sim(C_{ij}^t, C_{lj}^t) \tag{7}$$

where $d(C_{ij}^t, C_{lj}^t)$ is the fuzzy distance between $C_{ij}^t$ and $C_{lj}^t$, and $d(C_{ij}^t, C_{lj}^t) = (|Ex_{ij}^t - Ex_{lj}^t|, |En_{ij}^t - En_{lj}^t|, |He_{ij}^t - He_{lj}^t|)$; $\hat{s}(.)$ represents the total score of the cloud; $Sim_{il}^t$ is the average similarity between decision makers $e_i$ and $e_l$ with all $n$ alternatives.

This study clustered large groups using a traditional, direct method proposed by Zhou et al. [65]. The similarity matrix is denoted as $V^{tj} = [Sim_{il}^{tj}]_{m \times m}$, a symmetric matrix whose main diagonal element is 1. First, all elements of the upper triangle of matrix $V^{tj}$ (except the diagonal) were ranked from biggest to smallest and expressed as $o_1^t > o_2^t > \cdots > o_z^t$ (where $z \leq m(m-1)/2$). The grouping threshold is defined as $o_\tau^t (\tau = 1, 2, \cdots, z)$. If $Sim_{il}^{tj} > o_\tau^t$, decision makers $e_i$ and $e_l$ could be grouped together.

Secondly, the optimal grouping threshold is determined. The optimal grouping threshold $o_0^t$ is obtained by using threshold maximum rate of change method proposed by Zhou et al. [65]. Then, the change of grouping threshold is denoted by $U_i$:

$$U_i^t = \frac{o_{i-1}^t - o_i^t}{n_{i-1}^t - n_i^t} \tag{8}$$

where $i$ is the number of groups from smallest to largest; $n_{i-1}^t$ and $n_i^t$ are the number of members in the $(i-1)$th and $i$th groupings in round $t$, respectively; and $o_{i-1}^t$ and $o_i^t$ are the thresholds of the $(i-1)$th and $i$th groupings in round $t$, respectively. If

$$U_k^t = \max_i\{U_i^t\} \tag{9}$$

then the threshold of the $k$th group is considered to be optimal in round $t$, which means $o_0^t = o_k^t$.

Finally, the grouping results are decided. If $B_1$ and $B_2$ are two groups of the grouping threshold $o_\tau^t$ and $B_1 \cap B_2 \neq \varnothing$, then $B_1$ and $B_2$ are similar, which indicates that they can be combined into one group. The final classification is equivalent grouping of threshold $o_\tau^t$.

### 4.4. Determining the Weights of Decision Makers

Firstly, the weights of decision makers within the group are calculated. Authoritative decision makers pose a vital impact on the results in response to large-scale emergency decision-making issues. For example, the academician Zhong Nanshan made an accurate judgment on COVID-19 with his experiences and expertise that changed the layout of epidemic prevention and made the epidemic prevention measures more powerful. Hence, the highlighting of the importance of authoritative experts is critical. Our study measures the weights of decision makers on the basis of the trust relationship and the degree of trust by adopting the method proposed by Xiang et al. [66]. In the trust relationship, the importance of a certain expert equals the sum of the weighted trust of all other experts who trust him, which is expressed as

$$\mathrm{Confi}_{i_r}^{t(k+1)} = \sum_{l_r=1, l_r \neq i_r}^{\overline{m}_r^t} w_{l_r}^{tk} b_{l_r i_r}^t \ (i_r = 1, 2, \cdots m_r^t) \tag{10}$$

$$w_{i_r}^{t(k+1)} = \frac{\mathrm{Confi}_{i_r}^{t(k+1)}}{\sum\limits_{l_r=1}^{m_r^t} \mathrm{Confi}_{i_r}^{t(k+1)}} \tag{11}$$

where $\overline{m}_r^t$ refers to the total number of experts who have a trust relationship with the decision maker $i_r$ in the $r$th group of round $t$. $w_{l_r}^{tk}$ is the weight of decision maker $l_r$ in the $r$th group of round $t$ when the iteration round is $k$. $b_{l_r i_r}^t$ represents the trust degree of expert $l_r$ to expert $i_r$. The first iteration provides all the experts the same weight, that is, $w_{l_r}^{t1} = 1/m_r$. Then, we combine Equation (10) with (11) and apply the improved PageRank algorithm to constantly update the weights of experts. The iteration stops until $w_{i_r}^{t(k+1)} = w_{i_r}^{tk}$, and then $w_{i_r}^t = w_{i_r}^{t(k+1)}$.

Secondly, the weight of each group is determined. This study measures it, giving consideration to the similarity and group size within the group, namely, considering the density of data within the group, then

(1) The "quality" of information, attribute characteristics, can be presented as

$$\rho^{tr} = \frac{1}{(m_r)^2} \sum_{i_r=1}^{m_r} \sum_{l_r=1}^{m_r} Sim_{i_r l_r}^t \tag{12}$$

$$\xi_1^{tr} = \frac{\rho^{tr}}{\sum\limits_{r=1}^{q} \rho^{tr}} \tag{13}$$

where $\rho^{tr}$ denotes the average similarity in $r$th grouping in round $t$. The higher the value, the more intensive the data in the group. $m_r$ implies the number of decision makers in $r$th grouping in round $t$. $Sim_{i_r l_r}^t$ refers to the similarity between decision makers $l_r$ and $i_r$ in $r$th grouping, round $t$. $\xi_1^{tr}$ represents the "quality" weight of $r$th group in round $t$.

(2) The "quantity" of information, scale characteristics, is

$$\xi_2^{tr} = \frac{\beta^{tr}(m_r/m)^\alpha}{\sum\limits_{r=1}^{q} \beta^{tr}(m_r/m)^\alpha} \tag{14}$$

where $\xi_2^{tr}$ denotes the scale density weight, which has a positive correlation with the number of decision makers within the group. $\beta^{tr}$ refers to the density influencing factor, and $\alpha \in [-10, 10]$ can be decided according to the preference of the group.

Finally, we can obtain the grouping weights by multiplication normalization.

$$w^{tr} = \frac{\xi_1^{tr} \cdot \xi_2^{tr}}{\sum\limits_{r=1}^{q} \xi_1^{tr} \cdot \xi_2^{tr}} (r = 1, 2, \cdots, q) \tag{15}$$

According to Definition 5, we use the weights of experts within the group $(w_{i_r}^t)$ to calculate the cloud model $C_j^{tr}(Ex_j^{tr}, En_j^{tr}, He_j^{tr})$ of alternative $j$ in $r$th grouping in round $t$, as well as use the group weight $w^{tr}$ to evaluate the large group integrated cloud model $C_j^t(Ex_j^t, En_j^t, He_j^t)$ of alternative $j$, round $t$.

*4.5. Consensus Reaching Process*

In this section, we discuss the way in which to measure the consensus level of each expert and the way in which to adjust decision-making opinions as well as update trust. In large-group decision making, group consensus is an indispensable key that can ensure that multiple parties are satisfied with the decision outcomes. Moreover, the results could be more scientific, systematic and global if there is a reasonable consensus level. To this end, we constructed a consensus mechanism considering the update of trust.

(1) Determine the group consensus degree. The expected relative distance is used to measure the level of group consensus. The smaller the expected relative distance, the higher the consensus level; conversely, the lower. Then,

$$\hat{\theta}_{i_r}^{tj} = \frac{\left| Ex_{i_r}^{tj} - Ex_j^t \right|}{3(En_{i_r}^{tj} + En_j^t)} \tag{16}$$

$$p_{i_r}^{tj} = \begin{cases} 1 - \hat{\theta}_i^{tj}, 0 \le \hat{\theta}_{i_r}^{tj} < 1 \\ 0, 1 \le \hat{\theta}_{i_r}^{tj} \end{cases} \tag{17}$$

$$CL^{tj} = \sum_{r=1}^{q} \sum_{i_r=1}^{m_r} w^{tr} w_{i_r}^t p_{i_r}^{tj} \tag{18}$$

where $\hat{\theta}_{i_r}^{tj}$ refers to the relative distance between the decision maker $i_r$ and the group preference cloud model $C_j^t$ in $r$th grouping, round $t$. $p_{i_r}^{tj}$ is the consensus level of the decision maker $i_r$ in $r$th grouping in round $t$. $CL^{tj}$ indicates the group consensus level of alternative $j$ in round $t$.

Assuming the consensus threshold is $\delta$, if $CL^{tj} < \delta$, we adjust the opinions; otherwise, we directly rank the alternatives.

(2) Adjust opinions. In the process of opinion adjustment, experts tend to accept opinions whose trust or consensus reaches a certain confidence level. For this purpose, it is supposed that the confidence level of the expert $e_i$ on the trust degree is $\varepsilon_i$. If $\varepsilon_i < b_{il}^t$, then expert $e_i$ is willing to move closer to expert $e_l$'s opinions; otherwise, the opinions of expert $e_l$ will not be considered by expert $e_i$. Assuming that the consensus degree of expert $e_l$ is $p_l^{tj}$ about alternatives $j$ in round $t$, if $\delta < p_l^{tj}$, expert $e_i$ is willing to move closer to expert $e_l$'s opinions; otherwise, the opinions of expert $e_l$ will not be considered by expert $e_i$. In a large group of $m$ decision-makers, if the trust degree of $m_1^{ti}$ experts is greater than the confidence $\varepsilon_i$ of expert $i$ on the trust degree, and the consensus $p_l^{tj}$ of $m_2^{ti}$ experts is greater than the threshold $\delta$, then

$$\varphi_{il}^{tj} = \frac{b_{il}^t - \varepsilon_i}{\sum\limits_{l=1}^{m_1^{ti}} (b_{il}^t - \varepsilon_i)} \tag{19}$$

$$\phi_l^{tj} = \frac{p_l^{tj} - \delta}{\sum\limits_{l=1}^{m_2^{ti}} (p_l^{tj} - \delta)} \tag{20}$$

$$\gamma_{ij}^t = \begin{cases} \rho \sum\limits_{l=1}^{m_1^{ti}} \varphi_{il}^{tj} s_{lj}^t + (1 - \rho) \sum\limits_{l=1}^{m_2^{ti}} \phi_l^{tj} s_{lj}^t, m_1^{ti} \neq 0 \\ \sum\limits_{l=1}^{m_2^{ti}} \phi_l^{tj} s_{lj}^t, m_1^{ti} = 0 \end{cases} \tag{21}$$

$$s_{ij}^{(t+1)} = \begin{cases} s_{ij}^t + \pi_l^t(\gamma_{ij}^t - s_{ij}^t), s_{ij}^t \leq \gamma_{ij}^t \\ s_{ij}^t - \pi_l^t(s_{ij}^t - \gamma_{ij}^t), s_{ij}^t > \gamma_{ij}^t \end{cases} \tag{22}$$

where $\varphi_{il}^{tj}$ and $\phi_l^{tj}$ represent the willingness that the opinions of decision maker $i$ move closer to decision maker $l$ about alternatives $j$ in round $t$ from the perspective of trust and consensus, respectively. $\gamma_{ij}^t = [\gamma_{\theta_{ij}^t}^L, \gamma_{\theta_{ij}^t}^U]$ represents the adjustable upper limit of expert $i$'s opinion, taking both trust and consensus into consideration. $\pi_l^t$ represents expert $i$'s willingness to modify his opinion.

(3) Establish the trust updating mechanism. In the consensus-reaching process, the degree of trust among experts is susceptible to subtle changes due to several factors, including the degree of similarity of opinions, consensus level, and willingness to cooperate. Thus, we establish a trust-update mechanism to bring out the potential of trust in the decision-making process.

Assuming that $\eta_i = [\eta_i^L, \eta_i^U]$ is the confidence coefficient of expert $i$ on similarity, if $\eta_i^U < Sim_{il}^t$, expert $i$ will increase the trust in expert $l$ in round $(t+1)$; if $\eta_i^L > Sim_{il}^t$, expert $i$ will decrease the trust in expert $l$ in round $(t+1)$; if $\eta_i^L \leq Sim_{il}^t \leq \eta_i^U$, expert $i$'s trust in expert $l$ remains unchanged in round $(t+1)$. Then,

$$b_{il1}^{(t+1)} = \begin{cases} b_{il}^t(1 + \ln(\sqrt{1 + \overline{\sigma}_{l1}^t \chi_{11}}))^2, Sim_{il}^t \in (\eta_i^U, 1] \\ b_{il}^t \qquad\qquad , Sim_{il}^t \in [\eta_i^L, \eta_i^U] \\ b_{il}^t(1 - \ln(\sqrt{1 + \underline{\sigma}_{l1}^t \chi_{21}}))^2, Sim_{il}^t \in [0, \eta_i^L) \end{cases} \tag{23}$$

where $\overline{\sigma}_{l1}^t = \frac{Sim_{il}^t - \eta_i^U}{Sim_{il}^t}$ and $\underline{\sigma}_{l1}^t = \frac{\eta_i^L - Sim_{il}^t}{\eta_i^L}$ refer to the degree to which the similarity $Sim_{il}^t$ is superior to the upper limit $\eta_i^U$ or inferior to the lower limit $\eta_i^L$ of expert $i$'s similarity confidence, respectively. $\chi_{11}, \chi_{21}$ show the sensitivity coefficient of experts to similarity.

Assuming that the confidence of expert $i$ on the consensus degree is $\mu_i = [\mu_i^L, \mu_i^U]$, if $\mu_i^U < \overline{p}_l^t$, expert $i$ will increase the trust in expert $l$ in round $(t+1)$; if $\mu_i^L > \overline{p}_l^t$, expert $i$ will reduce it in expert $l$ in round $(t+1)$; if $\mu_i^L \leq \overline{p}_l^t \leq \mu_i^U$, expert $i$'s trust in expert $l$ stays unchanged in round $(t+1)$. Then,

$$b_{il2}^{(t+1)} = \begin{cases} b_{il}^t(1 + \ln(\sqrt{1 + \overline{\sigma}_{l2}^t \chi_{12}}))^2, \overline{p}_l^t \in (\mu_i^U, 1] \\ b_{il}^t \qquad\qquad , \overline{p}_l^t \in [\mu_i^L, \mu_i^U] \\ b_{il}^t(1 - \ln(\sqrt{1 + \underline{\sigma}_{l2}^t \chi_{22}}))^2, \overline{p}_l^t \in [0, \mu_i^L) \end{cases} \tag{24}$$

where $\overline{p}_l^t = \frac{1}{n^t} \sum\limits_{j=1}^n p_l^{tj}$ refers to the average consensus degree of expert $l$ on $n^t$ alternatives that have not reached consensus. $\overline{\sigma}_{l2}^t = \frac{\overline{p}_l^t - \mu_i^U}{\overline{p}_l^t}$ and $\underline{\sigma}_{l2}^t = \frac{\mu_i^L - \overline{p}_l^t}{\mu_i^L}$ refer to the degree to which average consensus degree $\overline{p}_l^t$ is superior to the upper limit $\mu_i^U$ or inferior to the lower limit $\mu_i^L$ of expert $i$'s consensus confidence, respectively. $\chi_{12}, \chi_{22}$ imply experts' sensitivity coefficient of consensus.

Assuming that $v_i = [v_i^L, v_i^U]$ is the confidence level of expert $i$ on the degree of opinions' revision, if $v_i^U < \pi_l^t$, expert $i$ will raise the trust in expert $l$ in round $(t+1)$; if $v_i^L > \pi_l^t$, expert $i$ will reduce trust in expert $l$ in round $(t+1)$; if $v_i^L \leq \pi_l^t \leq v_i^U$, expert $i$'s trust in expert $l$ stays in round $(t+1)$. Then,

$$b_{il3}^{(t+1)} = \begin{cases} b_{il}^t (1 + \ln(\sqrt{1 + \overline{\sigma}_{l3}^t \chi_{13}}))^2 & , \pi_l^t \in (v_i^U, 1] \\ b_{il}^t & , \pi_l^t \in [v_i^L, v_i^U] \\ b_{il}^t (1 - \ln(\sqrt{1 + \underline{\sigma}_{l3}^t \chi_{23}}))^2 & , \pi_l^t \in [0, v_i^L) \end{cases} \tag{25}$$

where $\pi_l^t$ implies the will of expert $l$ to modify his opinion. $\overline{\sigma}_{l3}^t = \frac{\pi_l^t - v_i^U}{\pi_l^t}$ and $\underline{\sigma}_{l3}^t = \frac{v_i^L - \pi_l^t}{v_i^L}$ refer to the degree to which the modification willingness $\pi_l^t$ is superior to the upper limit $v_i^U$ or inferior to the lower limit $v_i^L$ of expert $i$'s confidence, respectively. $\chi_{13}$, $\chi_{23}$ refer to experts' sensitivity coefficient of willingness in cooperation.

$$b_{il}^{(t+1)} = \lambda_1 b_{il1}^{(t+1)} + \lambda_2 b_{il2}^{(t+1)} + \lambda_3 b_{il3}^{(t+1)} \tag{26}$$

$$\lambda_1 + \lambda_2 + \lambda_3 = 1 \tag{27}$$

$b_{il}^{(t+1)j}$ is the updated degree of trust that integrates the similarity of opinions, the level of consensus with the degree of willingness in cooperation.

**Theorem 1.** *The consensus process proposed by this paper is convergent.*

**Proof.** Firstly, the trust degree of groups with high consensus will increase with $t$ according to Equation (26). Therefore, the consensus level of revision opinions $\sum\limits_{l=1}^{m_1^{ti}} \varphi_{il}^{tj} s_{lj}^t$ of $m_1^{ti}$ experts will also increase. Secondly, the consensus $p_l^{tj}$ of $m_2^{ti}$ experts is greater than the threshold $\delta$, so the consensus level of revision opinions $\sum\limits_{l=1}^{m_2^{ti}} \phi_l^{tj} s_{lj}^t$ of $m_2^{ti}$ experts must be greater than $\delta$. Then, the consensus level of $\gamma_{ij}^{t+1}$ will increase with $t$ according to Equation (21). Finally, the consensus level of $s_{ij}^{(t+1)}$ will be greater than $s_{ij}^t$ according to Equation (22), that is to say, the consensus process proposed by this paper is convergent. □

*4.6. The Sorting of the Alternatives*

Assuming that group opinions reach a consensus in round $h$, each alternative's comprehensive cloud model is $C_j^h (Ex_j^h, En_j^h, He_j^h)$. We define clouds $\overline{C}^h (\max\limits_j Ex_j^h, \min\limits_j En_j^h, \min\limits_j He_j^h)$ and $\underline{C}^h (\min\limits_j Ex_j^h, \max\limits_j En_j^h, \max\limits_j He_j^h)$ as the optimal and the worst cloud, respectively. Equations (2)–(8) show the way in which to to calculate the similarity $Sim_j^{h+}$ between the cloud model and the optimal cloud of the alternative $j$, and the similarity $Sim_j^{h-}$ between the cloud model and the worst cloud. Then, comprehensive similarity $Sim_j^h$ of each alternative is computed as follows:

$$Sim_j^h = \frac{Sim_j^{h+}}{Sim_j^{h+} + Sim_j^{h-}} \tag{28}$$

Apparently, the larger $Sim_j^h$ is, the better the alternative is.

## 5. An Illustrative Example

### 5.1. Background

A case of emergency plan selection for epidemic prevention and control illustrates the usefulness of the decision-making method proposed in our study. A certain university has several suspected cases of infectious diseases during the flu season. In order to prevent the situation from getting worse, the suspected cases were immediately quarantined. The school promptly organized eighteen experts ($m = 18$) in medical treatment and logistics overnight to study and judge the current situation, as well as to prepare to choose one of the four alternatives ($n = 4$) to deal with the current situation. Alternative $x_1$ needs to close the dormitory building of suspected patients and close contacts and disinfect the school every 12 h. People who enter and exit public areas are required to measure their temperature, register their information, and wear masks. Moreover, this plan advocates that all personnel do not go out unless necessary. Alternative $x_2$ adds a precautionary action on the basis of alternative $x_1$. It asks students at the suspected cases' college to take online lessons and be screened. Alternative $x_3$ adds several even stricter measures in alternative $x_1$. It lifts a ban on teachers' and students' movement and adopts a remote approach for teaching and learning. Furthermore, it requires that faculty, staff, and students are screened at least once, and everyone should stay until the official screening results are available. On the basis of plan $x_3$, alternative $x_4$ requires multiple rounds of screening for the whole school.

Hence, we assumed that the language phrase set was $S = \{s_0, s_1, s_2, s_3, s_4, s_5, s_6\}$. Table 2 shows uncertain language information for each alternative, given by experts who comprehensively considered the situation, control effect, mobilization intensity, cost, and so on. Table 3 represents the trust relationship among experts, while Table 4 shows the experts' confidence on the degree of trust, similarity, consensus, and opinion revision.

**Table 2.** Decision information of decision makers about alternatives.

| $x_j$ | $e_1$ | $e_2$ | $e_3$ | $e_4$ | $e_5$ | $e_6$ | $e_7$ | $e_8$ | $e_9$ |
|---|---|---|---|---|---|---|---|---|---|
| $x_1$ | $[s_0, s_3]$ | $[s_1, s_2]$ | $[s_2, s_5]$ | $[s_1, s_4]$ | $[s_1, s_3]$ | $[s_2, s_3]$ | $[s_2, s_5]$ | $[s_1, s_3]$ | $[s_0, s_2]$ |
| $x_2$ | $[s_1, s_4]$ | $[s_3, s_4]$ | $[s_2, s_3]$ | $[s_1, s_4]$ | $[s_1, s_5]$ | $[s_3, s_6]$ | $[s_2, s_4]$ | $[s_1, s_5]$ | $[s_1, s_3]$ |
| $x_3$ | $[s_3, s_5]$ | $[s_4, s_6]$ | $[s_4, s_5]$ | $[s_3, s_6]$ | $[s_5, s_5]$ | $[s_4, s_6]$ | $[s_2, s_3]$ | $[s_4, s_4]$ | $[s_2, s_6]$ |
| $x_4$ | $[s_3, s_4]$ | $[s_4, s_6]$ | $[s_3, s_4]$ | $[s_3, s_3]$ | $[s_2, s_5]$ | $[s_5, s_6]$ | $[s_3, s_3]$ | $[s_2, s_3]$ | $[s_2, s_4]$ |

| $x_j$ | $e_{10}$ | $e_{11}$ | $e_{12}$ | $e_{13}$ | $e_{14}$ | $e_{15}$ | $e_{16}$ | $e_{17}$ | $e_{18}$ |
|---|---|---|---|---|---|---|---|---|---|
| $x_1$ | $[s_0, s_3]$ | $[s_1, s_2]$ | $[s_0, s_4]$ | $[s_3, s_4]$ | $[s_1, s_4]$ | $[s_1, s_3]$ | $[s_3, s_6]$ | $[s_4, s_5]$ | $[s_1, s_3]$ |
| $x_2$ | $[s_3, s_4]$ | $[s_4, s_5]$ | $[s_4, s_6]$ | $[s_3, s_3]$ | $[s_2, s_5]$ | $[s_3, s_4]$ | $[s_1, s_3]$ | $[s_0, s_3]$ | $[s_1, s_4]$ |
| $x_3$ | $[s_4, s_5]$ | $[s_3, s_5]$ | $[s_1, s_4]$ | $[s_3, s_4]$ | $[s_2, s_4]$ | $[s_2, s_6]$ | $[s_4, s_4]$ | $[s_3, s_5]$ | $[s_5, s_6]$ |
| $x_4$ | $[s_3, s_5]$ | $[s_2, s_2]$ | $[s_1, s_5]$ | $[s_3, s_4]$ | $[s_3, s_4]$ | $[s_2, s_3]$ | $[s_1, s_4]$ | $[s_2, s_4]$ | $[s_4, s_6]$ |

**Table 3.** Trust information among decision makers.

| $e_i$ | $e_1$ | $e_2$ | $e_3$ | $e_4$ | $e_5$ | $e_6$ | $e_7$ | $e_8$ | $e_9$ | $e_{10}$ | $e_{11}$ | $e_{12}$ | $e_{13}$ | $e_{14}$ | $e_{15}$ | $e_{16}$ | $e_{17}$ | $e_{18}$ |
|---|---|---|---|---|---|---|---|---|---|---|---|---|---|---|---|---|---|---|
| $e_1$ | 1 | 0.6 | 0.4 | 0.5 | 0.7 | 0.9 | 0.6 | 0.8 | 0.4 | 0.5 | 0.8 | 0.3 | 0.7 | 0.9 | 0.4 | 0.5 | 0.6 | 0.8 |
| $e_2$ | 0.7 | 1 | 0.9 | 0.7 | 0.6 | 0.4 | 0.5 | 0.3 | 0.8 | 0.6 | 0.4 | 0.7 | 0.5 | 0.6 | 0.2 | 0.8 | 0.4 | 0.5 |
| $e_3$ | 0.4 | 0.6 | 1 | 0.5 | 0.4 | 0.8 | 0.7 | 0.9 | 0.6 | 0.4 | 0.5 | 0.6 | 0.3 | 0.4 | 0.2 | 0.5 | 0.9 | 0.7 |
| $e_4$ | 0.3 | 0.7 | 0.8 | 1 | 0.5 | 0.4 | 0.5 | 0.6 | 0.8 | 0.9 | 0.7 | 0.5 | 0.8 | 0.6 | 0.9 | 0.9 | 0.7 | 0.4 |
| $e_5$ | 0.6 | 0.5 | 0.6 | 0.4 | 1 | 0.6 | 0.3 | 0.2 | 0.8 | 0.5 | 0.9 | 0.7 | 0.5 | 0.6 | 0.4 | 0.5 | 0.4 | 0.8 |
| $e_6$ | 0.5 | 0.7 | 0.9 | 0.3 | 0.5 | 1 | 0.6 | 0.5 | 0.7 | 0.9 | 0.6 | 0.5 | 0.3 | 0.7 | 0.8 | 0.4 | 0.3 | 0.8 |
| $e_7$ | 0.7 | 0.6 | 0.3 | 0.8 | 0.9 | 0.4 | 1 | 0.7 | 0.6 | 0.5 | 0.8 | 0.9 | 0.4 | 0.3 | 0.6 | 0.8 | 0.7 | 0.6 |
| $e_8$ | 0.5 | 0.4 | 0.8 | 0.9 | 0.7 | 0.6 | 0.8 | 1 | 0.5 | 0.4 | 0.5 | 0.6 | 0.7 | 0.8 | 0.6 | 0.7 | 0.4 | 0.5 |
| $e_9$ | 0.8 | 0.6 | 0.8 | 0.9 | 0.7 | 0.6 | 0.5 | 0.7 | 1 | 0.5 | 0.6 | 0.7 | 0.8 | 0.6 | 0.4 | 0.9 | 0.5 | 0.7 |
| $e_{10}$ | 0.6 | 0.4 | 0.5 | 0.6 | 0.7 | 0.4 | 0.3 | 0.9 | 0.8 | 1 | 0.6 | 0.5 | 0.8 | 0.4 | 0.3 | 0.5 | 0.6 | 0.8 |
| $e_{11}$ | 0.5 | 0.3 | 0.5 | 0.7 | 0.9 | 0.6 | 0.4 | 0.8 | 0.2 | 0.6 | 1 | 0.9 | 0.4 | 0.5 | 0.6 | 0.2 | 0.7 | 0.8 |
| $e_{12}$ | 0.7 | 0.8 | 0.9 | 0.4 | 0.6 | 0.5 | 0.4 | 0.7 | 0.8 | 0.6 | 0.5 | 1 | 0.5 | 0.6 | 0.2 | 0.8 | 0.7 | 0.5 |
| $e_{13}$ | 0.5 | 0.7 | 0.6 | 0.8 | 0.4 | 0.6 | 0.3 | 0.5 | 0.4 | 0.9 | 0.6 | 0.6 | 1 | 0.8 | 0.7 | 0.7 | 0.7 | 0.8 |
| $e_{14}$ | 0.4 | 0.3 | 0.5 | 0.6 | 0.4 | 0.8 | 0.7 | 0.3 | 0.5 | 0.6 | 0.4 | 0.5 | 0.6 | 1 | 0.4 | 0.8 | 0.9 | 0.7 |
| $e_{15}$ | 0.8 | 0.5 | 0.6 | 0.7 | 0.8 | 0.4 | 0.5 | 0.6 | 0.3 | 0.7 | 0.8 | 0.5 | 0.4 | 0.6 | 1 | 0.7 | 0.8 | 0.6 |
| $e_{16}$ | 0.6 | 0.6 | 0.7 | 0.5 | 0.4 | 0.5 | 0.3 | 0.3 | 0.4 | 0.5 | 0.6 | 0.8 | 0.9 | 0.9 | 0.8 | 1 | 0.4 | 0.5 |
| $e_{17}$ | 0.7 | 0.5 | 0.8 | 0.6 | 0.7 | 0.8 | 0.8 | 0.5 | 0.6 | 0.4 | 0.7 | 0.6 | 0.3 | 0.4 | 0.2 | 0.8 | 1 | 0.9 |
| $e_{18}$ | 0.9 | 0.6 | 0.7 | 0.6 | 0.7 | 0.6 | 0.7 | 0.5 | 0.4 | 0.6 | 0.6 | 0.8 | 0.9 | 0.7 | 0.4 | 0.5 | 0.6 | 1 |

**Table 4.** Confidence level of experts regarding trust degree, similarity, consensus level, and cooperation willingness.

| $e_i$ | $e_1$ | $e_2$ | $e_3$ | $e_4$ | $e_5$ | $e_6$ | $e_7$ | $e_8$ | $e_9$ |
|---|---|---|---|---|---|---|---|---|---|
| $\varepsilon_i$ | 0.7 | 0.6 | 0.8 | 0.7 | 0.9 | 0.6 | 0.8 | 0.7 | 0.5 |
| $\eta_i^L$ | 0.75 | 0.66 | 0.86 | 0.55 | 0.81 | 0.74 | 0.59 | 0.69 | 0.76 |
| $\eta_i^U$ | 0.91 | 0.86 | 0.88 | 0.90 | 0.87 | 0.79 | 0.86 | 0.78 | 0.92 |
| $\mu_i^L$ | 0.85 | 0.87 | 0.84 | 0.81 | 0.88 | 0.86 | 0.84 | 0.83 | 0.88 |
| $\mu_i^U$ | 0.91 | 0.90 | 0.96 | 0.90 | 0.94 | 0.96 | 0.95 | 0.90 | 0.92 |
| $v_i^L$ | 0.45 | 0.4 | 0.42 | 0.35 | 0.35 | 0.30 | 0.35 | 0.36 | 0.35 |
| $v_i^U$ | 0.65 | 0.6 | 0.75 | 0.60 | 0.65 | 0.70 | 0.75 | 0.45 | 0.55 |

| $e_i$ | $e_{10}$ | $e_{11}$ | $e_{12}$ | $e_{13}$ | $e_{14}$ | $e_{15}$ | $e_{16}$ | $e_{17}$ | $e_{18}$ |
|---|---|---|---|---|---|---|---|---|---|
| $\varepsilon_i$ | 0.8 | 0.8 | 0.7 | 0.9 | 0.7 | 0.5 | 0.6 | 0.8 | 0.8 |
| $\eta_i^L$ | 0.68 | 0.86 | 0.85 | 0.79 | 0.90 | 0.78 | 0.59 | 0.77 | 0.87 |
| $\eta_i^U$ | 0.89 | 0.90 | 0.89 | 0.90 | 0.86 | 0.92 | 0.91 | 0.90 | 0.94 |
| $\mu_i^L$ | 0.86 | 0.85 | 0.87 | 0.90 | 0.90 | 0.82 | 0.86 | 0.88 | 0.87 |
| $\mu_i^U$ | 0.93 | 0.90 | 0.89 | 0.90 | 0.94 | 0.92 | 0.91 | 0.90 | 0.94 |
| $v_i^L$ | 0.40 | 0.40 | 0.45 | 0.40 | 0.50 | 0.40 | 0.30 | 0.60 | 0.42 |
| $v_i^U$ | 0.60 | 0.80 | 0.80 | 0.75 | 0.70 | 0.40 | 0.50 | 0.70 | 0.50 |

### 5.2. Calculation Steps

**Step 1.** Conversion of the cloud model and clustering of decision makers.

Assuming that the expert defines the universe of discourse as $[X_{\min}, X_{\max}] = [0, 100]$, the granular language can be transformed into the cloud matrix $R_{ij}^{1L}$ and $R_{ij}^{1U}$, with Formulas (2)–(5) in Section 3.2. Then, using Definition 4, $R_{ij}^{1L}$ and $R_{ij}^{1U}$ can be converted to $R_{ij}^1 = [C_{ij}^1(Ex_{ij}^1, En_{ij}^1, He_{ij}^1)]_{18 \times 4}$, as shown in Table 5. Moreover, according to Formulas (8) and (9), the first-round clustering threshold was calculated, $o_0^1 = 0.885$, and the decision-making group was divided into six ($r = 6$) groups: $E_1 = \{1, 3, 4, 10, 11, 13, 14\}$, $E_2 = \{5, 7, 8, 15, 16, 17, 18\}$, $E_3 = \{2\}$, $E_4 = \{6\}$, $E_5 = \{9\}$, and $E_6 = \{12\}$.

**Table 5.** Comprehensive cloud in the first round.

| $x_j$ | $e_1$ | $e_2$ | $e_3$ | $e_4$ | $e_5$ | $e_6$ |
|---|---|---|---|---|---|---|
| $x_1$ | (38.67, 41.34, 0.36) | (37.63, 05.18, 0.26) | (50.60, 09.10, 0.26) | (49.40, 09.10, 0.26) | (41.06, 23.98, 0.23) | (45.33, 15.06, 0.18) |
| $x_2$ | (49.40, 09.10, 0.26) | (54.67, 15.06, 0.18) | (45.33, 15.06, 0.18) | (49.40, 09.10, 0.26) | (50.00, 09.30, 0.30) | (61.33, 41.34, 0.36) |
| $x_3$ | (58.94, 23.98, 0.23) | (73.56, 37.27, 0.37) | (62.37, 05.18, 0.26) | (61.33, 41.34, 0.36) | (77.90, 00.00, 0.30) | (73.56, 37.27, 0.37) |
| $x_4$ | (54.67, 15.06, 0.18) | (73.56, 37.27, 0.37) | (54.67, 15.06, 0.18) | (50.00, 00.00, 0.14) | (50.60, 09.10, 0.26) | (86.37, 33.00, 0.40) |

| $x_j$ | $e_7$ | $e_8$ | $e_9$ | $e_{10}$ | $e_{11}$ | $e_{12}$ |
|---|---|---|---|---|---|---|
| $x_1$ | (50.60, 09.10, 0.26) | (41.06, 23.98, 0.23) | (26.44, 37.27, 0.37) | (38.67, 41.34, 0.35) | (37.63, 05.18, 0.26) | (54.56, 18.18, 0.37) |
| $x_2$ | (50.00, 03.92, 0.22) | (50.00, 09.30, 0.30) | (41.06, 23.98, 0.23) | (54.67, 15.06, 0.18) | (62.37, 05.18, 0.26) | (73.56, 37.27, 0.37) |
| $x_3$ | (45.33, 15.05, 0.18) | (61.77, 00.00, 0.22) | (45.47, 18.18, 0.37) | (62.37, 05.18, 0.26) | (58.94, 23.98, 0.23) | (49.40, 09.10, 0.26) |
| $x_4$ | (50.00, 00.00, 0.14) | (45.33, 15.06, 0.18) | (50.00, 03.92, 0.22) | (58.94, 23.98, 0.23) | (38.23, 00.00, 0.22) | (50.00, 09.30, 0.30) |

| $x_j$ | $e_{13}$ | $e_{14}$ | $e_{15}$ | $e_{16}$ | $e_{17}$ | $e_{18}$ |
|---|---|---|---|---|---|---|
| $x_1$ | (54.67, 15.06, 0.18) | (49.40, 09.10, 0.26) | (41.06, 23.98, 0.23) | (61.33, 41.34, 0.36) | (62.37, 05.18, 0.26) | (41.06, 23.98, 0.23) |
| $x_2$ | (50.00, 00.00, 0.14) | (50.60, 09.10, 0.26) | (54.67, 15.06, 0.18) | (41.06, 23.98, 0.23) | (38.67, 41.34, 0.35) | (49.40, 09.10, 0.26) |
| $x_3$ | (54.67, 15.05, 0.18) | (50.00, 03.92, 0.22) | (45.47, 18.18, 0.37) | (61.77, 00.00, 0.22) | (58.94, 23.98, 0.23) | (86.37, 33.00, 0.40) |
| $x_4$ | (54.67, 15.06, 0.18) | (50.60, 09.10, 0.26) | (45.33, 15.06, 0.18) | (49.40, 09.10, 0.26) | (50.00, 03.92, 0.22) | (73.56, 37.27, 0.37) |

**Step 2.** Determine the weights of decision makers and groups.

According to the above, the trust relationship was divided into six groups. With Formulas (10) and (11), the weights of decision makers in each group were

$$w_{i_1}^1 = [0.1147, 0.1375, 0.1513, 0.1591, 0.1456, 0.1475, 0.1443] w_{i_3}^1 = w_{i_4}^1 = w_{i_5}^1 = w_{i_6}^1 = 1$$

$$w_{i_2}^1 = [0.1665, 0.1370, 0.1158, 0.1263, 0.1593, 0.1358, 0.1593]$$

The weight vector of each group was $w^{1r} = [0.3792, 0.3859, 0.0587, 0.0587, 0.0587, 0.0587]$, using Formulas (12)–(15) and $\alpha = 1$ in Formula (14). Therefore, according to the weights of decision makers and the groups, each alternative's comprehensive cloud $C_j^{1r}(Ex_j^{1r}, En_j^{1r}, He_j^{1r})$ in one group was calculated using Definition 5:

$$C_j^1 = [(45.66, 23.62, 0.286), (51.53, 20.50, 0.261), (60.92, 22.00, 0.292), (55.16, 15.00, 0.255)]$$

**Step 3.** Consensus-reaching process.

(1) Calculate the consensus degree. Assuming that the consensus degree threshold was $\delta = 0.95$, the consensus level of all the alternatives was obtained using Formulas (16)–(18). It can be seen that only the alternative $x_2$ reached consensus. The decision information of other alternatives needed to be adjusted.

(2) Determine adjustment of opinions. The adjustment of opinions was determined by trust and the consensus degree. Supposing $\rho = 0.4$, the upper limit of the adjustment was calculated according to Formulas (19)–(21), as shown in Table 6. Then, the modification willingness of 18 decision makers, using the rand function, were

$$\pi_i = [0.176, 0.723, 0.153, 0.341, 0.607, 0.192, 0.738, 0.243, 0.917, 0.269, 0.766, 0.189, 0.288,$$
$$0.091, 0.576, 0.683, 0.547, 0.426]$$

**Table 6.** The upper bound of the first round of adjustment opinions.

| $x_j$ | $e_1$ | $e_2$ | $e_3$ | $e_4$ | $e_5$ | $e_6$ |
|---|---|---|---|---|---|---|
| $x_1$ | $[s_{1.026}, s_{3.124}]$ | $[s_{1.152}, s_{3.312}]$ | $[s_{1.566}, s_{3.684}]$ | $[s_{1.200}, s_{3.450}]$ | $[s_{1.066}, s_{3.084}]$ | $[s_{1.137}, s_{3.201}]$ |
| $x_3$ | $[s_{3.404}, s_{4.946}]$ | $[s_{3.426}, s_{5.060}]$ | $[s_{3.584}, s_{4.846}]$ | $[s_{3.350}, s_{5.046}]$ | $[s_{4.084}, s_{4.946}]$ | $[s_{3.542}, s_{5.158}]$ |
| $x_4$ | $[s_{3.040}, s_{4.122}]$ | $[s_{2.886}, s_{4.150}]$ | $[s_{2.800}, s_{3.822}]$ | $[s_{2.767}, s_{3.822}]$ | $[s_{2.600}, s_{4.322}]$ | $[s_{3.165}, s_{4.298}]$ |

| $x_j$ | $e_7$ | $e_8$ | $e_9$ | $e_{10}$ | $e_{11}$ | $e_{12}$ |
|---|---|---|---|---|---|---|
| $x_1$ | $[s_{1.166}, s_{3.584}]$ | $[s_{1.166}, s_{3.434}]$ | $[s_{1.161}, s_{3.331}]$ | $[s_{0.800}, s_{3.084}]$ | $[s_{0.966}, s_{2.984}]$ | $[s_{1.066}, s_{3.484}]$ |
| $x_3$ | $[s_{3.084}, s_{4.446}]$ | $[s_{3.384}, s_{4.746}]$ | $[s_{3.389}, s_{4.958}]$ | $[s_{3.684}, s_{4.813}]$ | $[s_{3.284}, s_{4.846}]$ | $[s_{3.134}, s_{4.846}]$ |
| $x_4$ | $[s_{2.700}, s_{3.922}]$ | $[s_{2.800}, s_{3.672}]$ | $[s_{2.788}, s_{3.981}]$ | $[s_{2.867}, s_{4.055}]$ | $[s_{2.500}, s_{3.722}]$ | $[s_{2.600}, s_{4.172}]$ |

| $x_j$ | $e_{13}$ | $e_{14}$ | $e_{15}$ | $e_{16}$ | $e_{17}$ | $e_{18}$ |
|---|---|---|---|---|---|---|
| $x_1$ | $[s_{1.866}, s_{3.484}]$ | $[s_{1.581}, s_{3.655}]$ | $[s_{1.200}, s_{3.291}]$ | $[s_{1.413}, s_{3.670}]$ | $[s_{1.866}, s_{3.617}]$ | $[s_{1.166}, s_{3.184}]$ |
| $x_3$ | $[s_{3.350}, s_{5.046}]$ | $[s_{4.084}, s_{4.946}]$ | $[s_{3.542}, s_{5.158}]$ | $[s_{3.084}, s_{4.446}]$ | $[s_{3.384}, s_{4.746}]$ | $[s_{3.389}, s_{4.958}]$ |
| $x_4$ | $[s_{3.000}, s_{3.922}]$ | $[s_{2.714}, s_{4.208}]$ | $[s_{2.719}, s_{3.833}]$ | $[s_{2.547}, s_{4.002}]$ | $[s_{2.867}, s_{4.188}]$ | $[s_{3.200}, s_{4.322}]$ |

Table 7 shows the opinions in the second round with Formula (22).

**Table 7.** The second-round decision information of decision makers about alternatives.

| $x_j$ | $e_1$ | $e_2$ | $e_3$ | $e_4$ | $e_5$ | $e_6$ |
|---|---|---|---|---|---|---|
| $x_1$ | $[s_{0.181}, s_{3.022}]$ | $[s_{1.110}, s_{2.949}]$ | $[s_{1.934}, s_{4.799}]$ | $[s_{1.068}, s_{3.813}]$ | $[s_{1.040}, s_{3.051}]$ | $[s_{1.834}, s_{3.093}]$ |
| $x_2$ | $[s_1, s_4]$ | $[s_3, s_4]$ | $[s_2, s_3]$ | $[s_1, s_4]$ | $[s_1, s_5]$ | $[s_3, s_6]$ |
| $x_3$ | $[s_{3.071}, s_{4.991}]$ | $[s_{3.585}, s_{5.321}]$ | $[s_{3.936}, s_{4.976}]$ | $[s_{3.119}, s_{5.675}]$ | $[s_{4.444}, s_{4.967}]$ | $[s_{3.912}, s_{5.838}]$ |
| $x_4$ | $[s_{3.007}, s_{4.021}]$ | $[s_{3.194}, s_{4.663}]$ | $[s_{2.969}, s_{3.973}]$ | $[s_{2.920}, s_{3.280}]$ | $[s_{2.364}, s_{4.588}]$ | $[s_{4.648}, s_{5.673}]$ |

| $x_j$ | $e_7$ | $e_8$ | $e_9$ | $e_{10}$ | $e_{11}$ | $e_{12}$ |
|---|---|---|---|---|---|---|
| $x_1$ | $[s_{1.385}, s_{3.955}]$ | $[s_{1.040}, s_{3.105}]$ | $[s_{1.064}, s_{3.220}]$ | $[s_{0.215}, s_{3.023}]$ | $[s_{0.974}, s_{2.754}]$ | $[s_{0.202}, s_{3.902}]$ |
| $x_2$ | $[s_2, s_4]$ | $[s_1, s_5]$ | $[s_1, s_3]$ | $[s_3, s_4]$ | $[s_4, s_5]$ | $[s_4, s_6]$ |
| $x_3$ | $[s_{2..800}, s_{4.067}]$ | $[s_{3.850}, s_{4.181}]$ | $[s_{3.274}, s_{5.044}]$ | $[s_{3.915}, s_{4.950}]$ | $[s_{3.217}, s_{4.882}]$ | $[s_{1.403}, s_{4.160}]$ |
| $x_4$ | $[s_{2.779}, s_{3.680}]$ | $[s_{2.194}, s_{3.163}]$ | $[s_{2.723}, s_{3.982}]$ | $[s_{2.964}, s_{4.746}]$ | $[s_{2.383}, s_{3.319}]$ | $[s_{1.302}, s_{4.843}]$ |

| $x_j$ | $e_{13}$ | $e_{14}$ | $e_{15}$ | $e_{16}$ | $e_{17}$ | $e_{18}$ |
|---|---|---|---|---|---|---|
| $x_1$ | $[s_{2.674}, s_{3.851}]$ | $[s_{1.053}, s_{3.969}]$ | $[s_{1.115}, s_{3.168}]$ | $[s_{1.916}, s_{4.409}]$ | $[s_{2.833}, s_{4.244}]$ | $[s_{1.071}, s_{3.078}]$ |
| $x_2$ | $[s_3, s_3]$ | $[s_2, s_5]$ | $[s_3, s_4]$ | $[s_1, s_3]$ | $[s_0, s_3]$ | $[s_1, s_4]$ |
| $x_3$ | $[s_{3.082}, s_{4.157}]$ | $[s_{2.112}, s_{4.070}]$ | $[s_{2.808}, s_{5.427}]$ | $[s_{3.438}, s_{4.464}]$ | $[s_{3.301}, s_{5.043}]$ | $[s_{4.439}, s_{5.594}]$ |
| $x_4$ | $[s_{3.000}, s_{3.977}]$ | $[s_{2.065}, s_{4.928}]$ | $[s_{2.414}, s_{3.480}]$ | $[s_{2.056}, s_{4.001}]$ | $[s_{2.474}, s_{4.103}]$ | $[s_{3.659}, s_{5.285}]$ |

(3) Updating of the trust level.

Since decision makers have different sensitivities to the better or worse of the attribute state, this study charted the value change of different sensitivity coefficients. Judging from all evidence offered by Figures 2 and 3, we may safely draw the conclusion that the value range of the sensitivity coefficient will be more reasonable if there is $\chi_{11}, \chi_{12}, \chi_{13} \in [1.5, 2.5]$ and $\chi_{21}, \chi_{22}, \chi_{23} \in [0.6, 1]$. It was supposed that the similarity of opinions, consensus level, and cooperation willingness had the same influence on trust, namely, sensitivity coefficients $\chi_{21} = \chi_{22} = \chi_{23} = 1$ and $\lambda_1 = \lambda_2 = \lambda_3 = 1/3$. Therefore, the trust degree in the second round was obtained with Formulas (23)–(27). Due to space limitations, the updated trust was omitted. We then repeated steps 1–3 until the consensus level was reached.

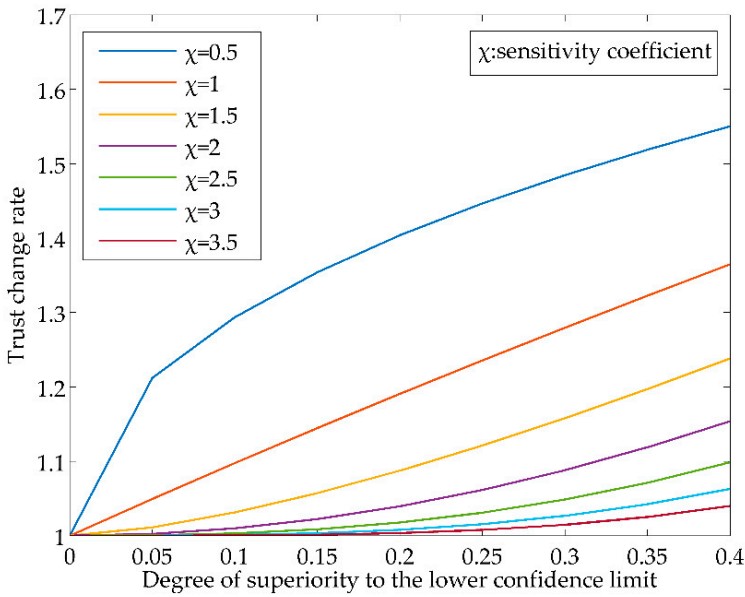

**Figure 2.** The influence of sensitivity coefficient on trust increase.

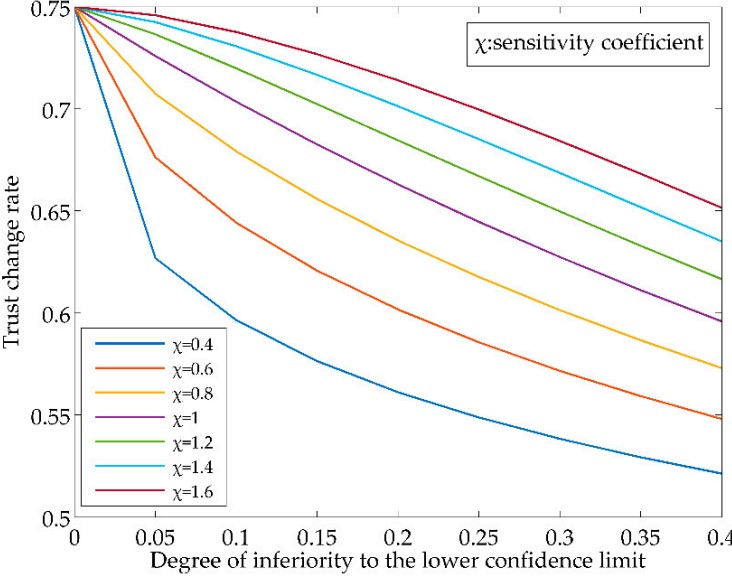

**Figure 3.** The influence of sensitivity coefficient on trust reduction.

**Step 4.** Sorting of the alternatives.

Table 8 shows the decision information after four rounds of adjustment. The consensus level of the alternative was then $CL^{5j} = [0.9775, 0.9726, 0.9501, 0.9626]$. Thus, the decision makers reached a consensus in the fifth round. The comprehensive cloud of the alternatives was

$$C_j^5 = [(45.61, 13.151, 0.249), (50.91, 10.266, 0.242), (56.58, 8.423, 0.248), (50.58, 3.415, 0.194)].$$

Furthermore, the optimal cloud and the worst cloud were obtained as $\overline{C}^5(56.582, 3.415, 0.194)$ and $\underline{C}^5(48.605, 13.151, 0.249)$, respectively. The comprehensive similarity of alternatives was obtained as $Sim_j^5 = [0.4896, 0.5127, 0.5424, 0.5029]$ according to Formula (28). That is to say, alternative $x_3$ is the optimum and the selection.

**Table 8.** The fifth-round decision information of decision makers about alternatives.

| $x_j$ | $e_1$ | $e_2$ | $e_3$ | $e_4$ | $e_5$ | $e_6$ |
|---|---|---|---|---|---|---|
| $x_1$ | $[s_{0.181}, s_{3.022}]$ | $[s_{1.110}, s_{2.949}]$ | $[s_{1.934}, s_{4.799}]$ | $[s_{1.068}, s_{3.813}]$ | $[s_{1.040}, s_{3.051}]$ | $[s_{1.834}, s_{3.093}]$ |
| $x_2$ | $[s_{1.142}, s_{4.044}]$ | $[s_{2.170}, s_{3.952}]$ | $[s_{1.944}, s_{3.115}]$ | $[s_{1.253}, s_{3.998}]$ | $[s_{1.412}, s_{4.728}]$ | $[s_{2.826}, s_{5.681}]$ |
| $x_3$ | $[s_{3.238}, s_{5.016}]$ | $[s_{3.494}, s_{5.000}]$ | $[s_{3.801}, s_{4.995}]$ | $[s_{3.350}, s_{5.232}]$ | $[s_{3.728}, s_{5.000}]$ | $[s_{3.764}, s_{5.508}]$ |
| $x_4$ | $[s_{2.925}, s_{3.873}]$ | $[s_{2.647}, s_{3.612}]$ | $[s_{2.885}, s_{3.837}]$ | $[s_{2.753}, s_{3.461}]$ | $[s_{2.575}, s_{3.732}]$ | $[s_{3.847}, s_{4.812}]$ |

| $x_j$ | $e_7$ | $e_8$ | $e_9$ | $e_{10}$ | $e_{11}$ | $e_{12}$ |
|---|---|---|---|---|---|---|
| $x_1$ | $[s_{1.385}, s_{3.955}]$ | $[s_{1.040}, s_{3.105}]$ | $[s_{1.064}, s_{3.220}]$ | $[s_{0.215}, s_{3.023}]$ | $[s_{0.974}, s_{2.754}]$ | $[s_{0.202}, s_{3.902}]$ |
| $x_2$ | $[s_{1.826}, s_{4.202}]$ | $[s_{1.150}, s_{4.823}]$ | $[s_{1.646}, s_{3.936}]$ | $[s_{2.780}, s_{4.024}]$ | $[s_{2.703}, s_{4.656}]$ | $[s_{3.670}, s_{5.703}]$ |
| $x_3$ | $[s_{3.523}, s_{4.912}]$ | $[s_{3.652}, s_{4.600}]$ | $[s_{3.501}, s_{4.986}]$ | $[s_{3.742}, s_{4.980}]$ | $[s_{3.567}, s_{4.990}]$ | $[s_{2.221}, s_{4.494}]$ |
| $x_4$ | $[s_{2.607}, s_{3.596}]$ | $[s_{2.453}, s_{3.325}]$ | $[s_{2.594}, s_{3.597}]$ | $[s_{2.849}, s_{4.193}]$ | $[s_{2.569}, s_{3.564}]$ | $[s_{1.852}, s_{4.337}]$ |

| $x_j$ | $e_{13}$ | $e_{14}$ | $e_{15}$ | $e_{16}$ | $e_{17}$ | $e_{18}$ |
|---|---|---|---|---|---|---|
| $x_1$ | $[s_{2.674}, s_{3.851}]$ | $[s_{1.053}, s_{3.969}]$ | $[s_{1.115}, s_{3.168}]$ | $[s_{1.916}, s_{4.409}]$ | $[s_{2.833}, s_{4.244}]$ | $[s_{1.071}, s_{3.078}]$ |
| $x_2$ | $[s_{2.801}, s_{3.216}]$ | $[s_{1.961}, s_{4.922}]$ | $[s_{2.357}, s_{4.101}]$ | $[s_{1.699}, s_{3.627}]$ | $[s_{0.624}, s_{3.417}]$ | $[s_{1.217}, s_{4.065}]$ |
| $x_3$ | $[s_{3.267}, s_{4.535}]$ | $[s_{2.390}, s_{4.262}]$ | $[s_{3.454}, s_{5.050}]$ | $[s_{3.412}, s_{4.888}]$ | $[s_{3.460}, s_{5.025}]$ | $[s_{3.888}, s_{5.252}]$ |
| $x_4$ | $[s_{1.852}, s_{4.337}]$ | $[s_{2.867}, s_{3.747}]$ | $[s_{2.190}, s_{4.654}]$ | $[s_{2.615}, s_{3.582}]$ | $[s_{2.585}, s_{3.607}]$ | $[s_{3.077}, s_{4.206}]$ |

### 5.3. Comparative Analysis

If the consensus process did not consider the trust update, namely, the calculation process does not conduct the operation of Equations (23)–(26), it turned out that the consensus level of the decision-making opinion was $\overline{CL}^{6j} = [0.9703, 0.9702, 0.9506, 0.9563]$ after five rounds of revisions to reach the threshold level. However, with the method proposed in our study, it took only four rounds of adjustment to come to the threshold.

Moreover, the comprehensive cloud of the sixth round was $C_j^6 = [(48.40, 11.089, 0.432), (50.94, 8.921, 0.427), (55.28, 5.945, 0.432), (50.67, 2.884, 0.388)]$, and the comprehensive similarity of the alternatives was $Sim_j^6 = [0.4907, 0.5214, 0.5254, 0.5032]$, which means that alternative $x_3$ was the best. Compared with the above analysis, as shown in Figure 4, it was apparent that the results of alternative $x_3$ and alternative $x_2$ were almost the same when trust update was not considered; conversely, plan $x_3$ became more distinguished when considering the trust-updating mechanism.

Further analysis of Figure 5 shows that the no-trust-updating method had a lower difference of weighting among decision makers than the trust-updating mechanism. That is, considering trust update can speed up the consensus-reaching process, lower the costs, and guarantee that the opinion leaders could play a significant role in the decision-making process as their weights were clearly differentiated. The analysis we made confirmed the view that the consensus mechanism constructed in this study was effective.

The consensus degree of the alternatives at each opinion interaction stage is shown in Figure 6. As shown in Figure 6a, alternative 2 reached consensus in the first round, so its opinions were not updated in the second and third rounds. Before the fourth round of the interaction, the consensus level of alternative 2 was lower than the threshold. The reason was that the weight of each decision maker changed greatly, so the decision maker needed to adjust the opinions of alternative 2 in the fourth round. For the same reason, the consensus level of alternative 2 decreased at the beginning of the fifth round, as shown in Figure 6b.

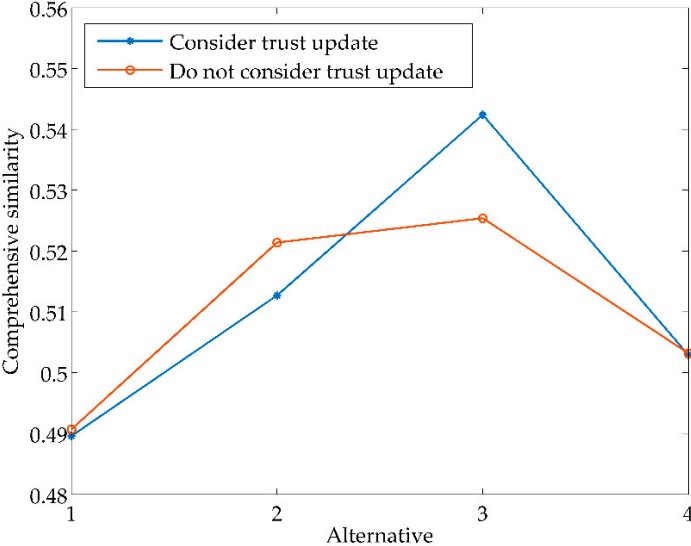

**Figure 4.** The influence of trust updating on decision results.

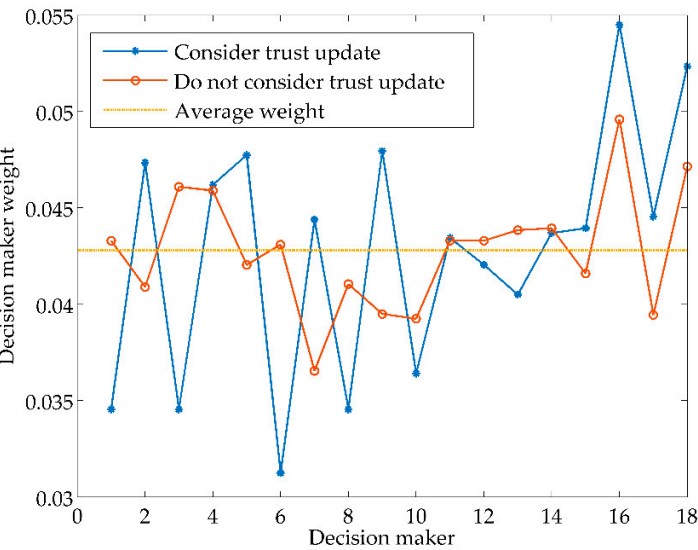

**Figure 5.** The influence of trust updating on the weight of decision makers.

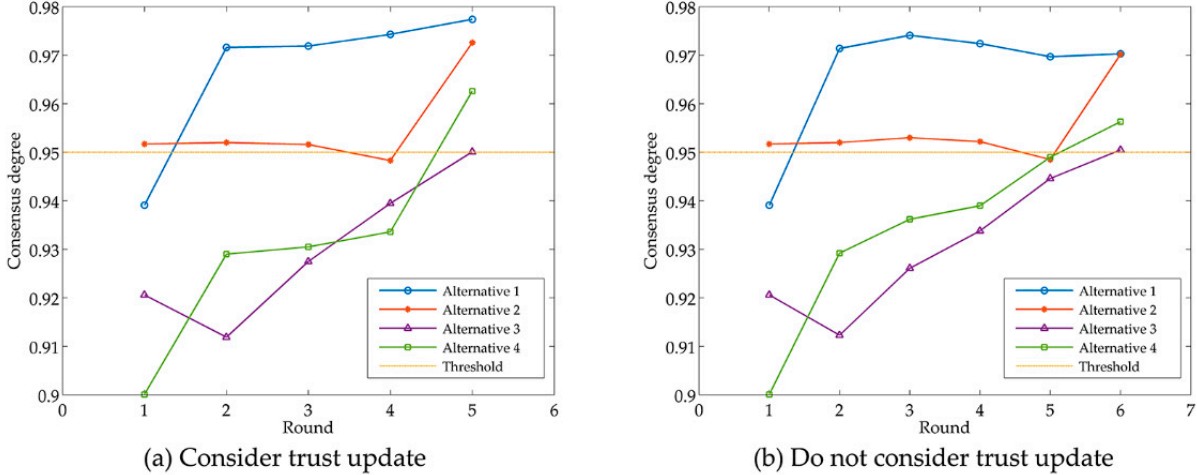

**Figure 6.** Consensus degree of the alternatives at each opinion interaction stage.

Clustering at each opinion interaction stage is shown in Figure 7. As shown in Figure 7, the grouping of decision makers was different at each stage. Decision makers were finally clustered into four groups when considering trust update, but five groups when not considering it. In addition, it was found that there were always some groups that only had one or two members. For example, decision maker 12 was always clustered into a group, only on its own. The reason is that the confidence of decision maker 12 in trust, similarity, and consensus was high, which indicates that it was not easy to update their trust degree. In addition, the cooperation willingness of decision maker 12 was only 0.189, which expresses that it was not easy for them to accept others' opinions. If the opinion of decision maker 12 was rejected after the fourth round of consensus with updating information of trust, the consensus level of each alternative was $\overline{CL}^{4j} = [0.9726, 0.9564, 0.9439, 0.9284]$. The original consensus level was $CL^{4j} = [0.9743, 0.9483, 0.9395, 0.9336]$. That is to say, the opinions of decision maker 12 had a negative impact on the consensus of alternatives 2 and 3 and a positive impact on alternatives 1 and 4. Decision-making members may not cooperate or have low willingness to cooperate in LGEDM, and their opinions exerted an important effect on reaching a consensus. Therefore, in the decision-making process, if the opinions of these groups can be identified in advance, their impact will be avoided, and consensus will be reached more effectively.

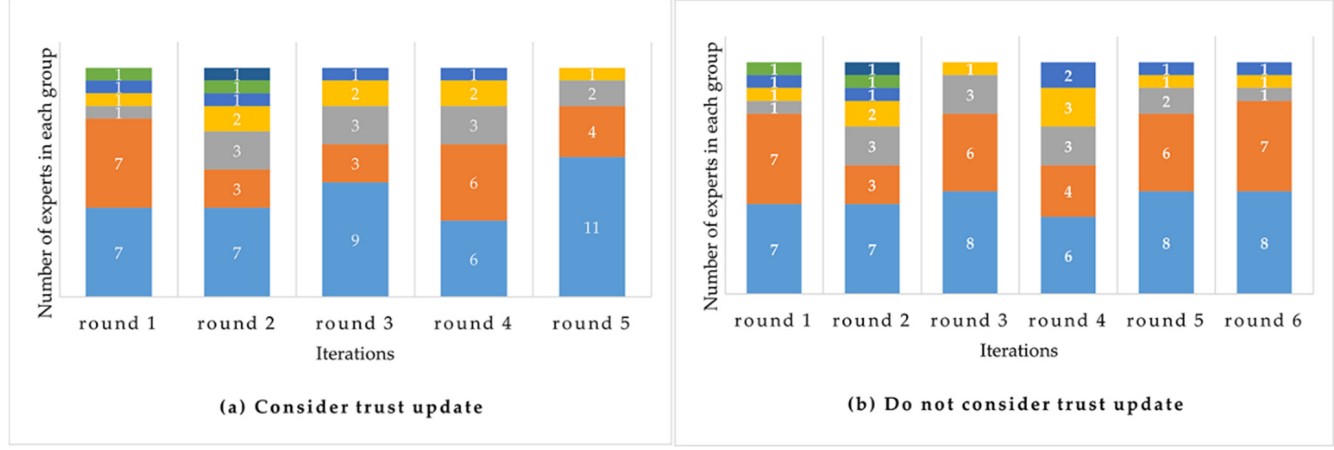

**Figure 7.** Clustering at each opinion interaction stage.

Convergence and divergence of alternative information is shown in Figure 8. After reaching a consensus, the center of gravity of a cloud droplet of alternative 3 was the largest, and that of alternative 1 was the smallest. Namely, alternative 3 had the largest expectation, while alternative 1 had the smallest. The cloud droplet span of alternative 1 was the largest, and that of alternative 4 was the smallest. Namely, alternative 1 had the largest uncertainty, while alternative 4 had the smallest. In addition, the cloud droplet thickness of alternatives 3 and 4 were higher than alternatives 1 and 2. Hence, according to the gravity center, span, and thickness of cloud droplets, alternative 3 was the best.

In order to compare the alternatives more specifically, the concept of optimal cloud was introduced. As shown in Figure 9, after the interaction of opinions, the cloud droplet span of each alternative was significantly reduced. That is to say, the astringency was significantly enhanced, which means the certainty of decision information increased. It is easy to know that the cloud droplets were quite similar whether considering the trust update or not. This shows that trust update had little impact on the decision results, but it could effectively speed up the consensus-reaching process. It can be seen from Figure 9c that alternative 3 had the highest coincidence with the optimal cloud, which means it had the highest similarity, namely, it was the optimal choice.

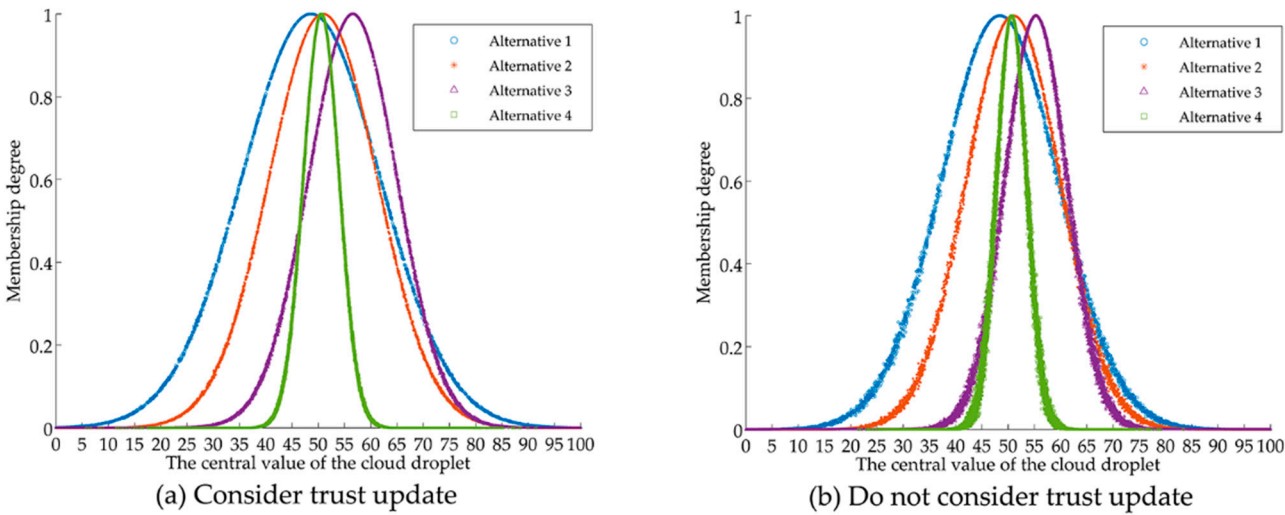

**Figure 8.** Integrated cloud of alternatives after reaching a consensus.

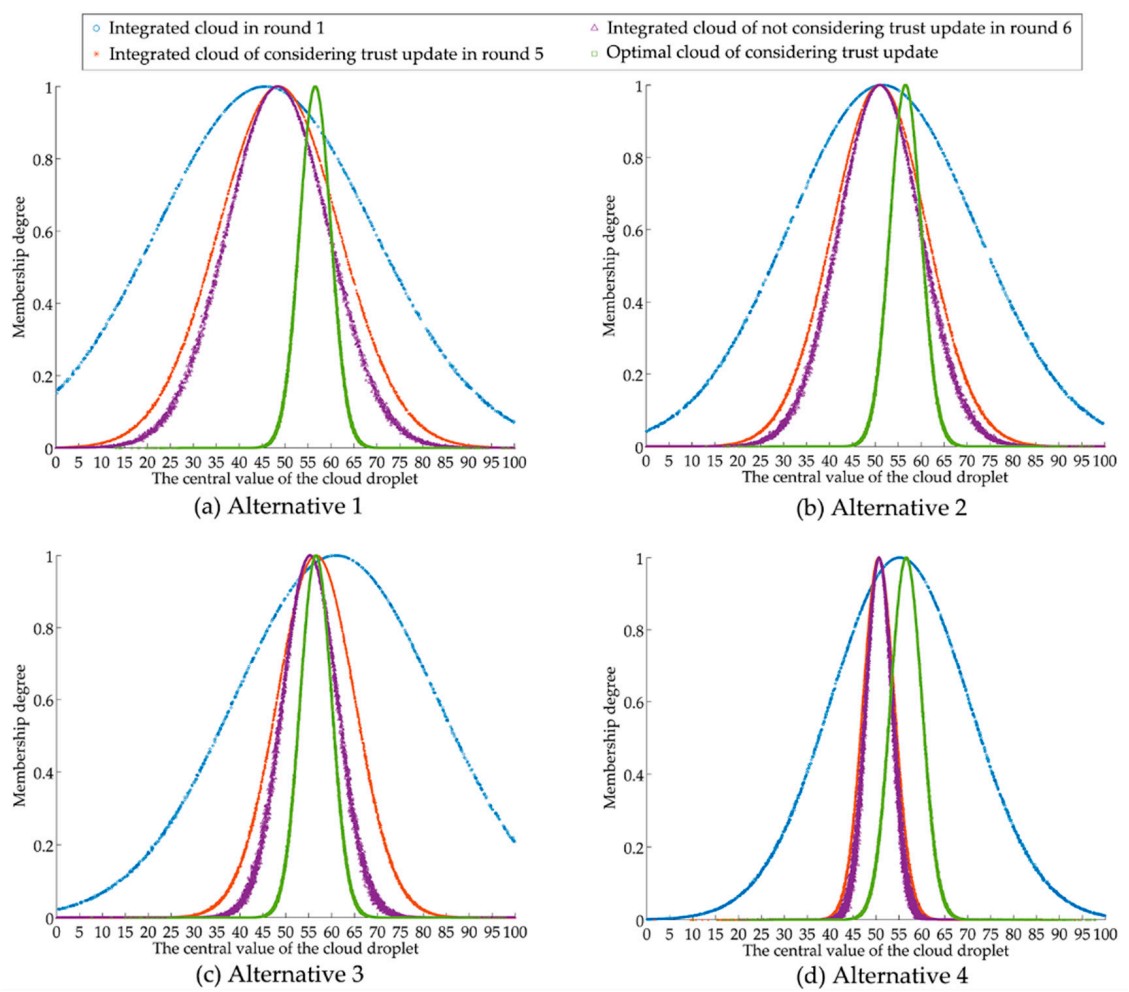

**Figure 9.** Integrated cloud of alternatives at each opinion interaction stage.

## 6. Conclusions

This research proposed a novel dynamic, trust-driven cloud similarity decision-making method based on large-group consensus and uncertain linguistic information to effectively solve emergency decision-making problems. Our major findings are as follows: (1) In

order to take full advantage of the randomness and fuzziness of continuous language, we transformed it into cloud models in the information aggregation process, which can minimize the loss of information. (2) From the perspective of decision outcomes, taking updated information of trust into consideration would pose an effect on differential validity of results, but would have less impact on the choice of alternatives. Conversely, in terms of the decision-making process, it turns out that considering a scheme of trust updating is essential for a cost-reducing decision because the scheme could reduce the number of opinions exchanged and accelerate the consensus-reaching process. (3) Experts' revising their opinions consider not only similarity but also consensus in real life. Therefore, the proposed opinion interaction mechanism considering consensus level and trust degree is reasonable and feasible.

As with any other research work, this study suffers a couple of limitations. One limitation concerns the decision-making group. In this study, the way in which to identify groups with low willingness to cooperate in clustering and the way in which to avoid the negative effects of non-cooperative groups was not explored. Another limitation concerns the exploration of relationship between decision makers' confidence and willingness in cooperation, which may lead to further research that could conduct an in-depth study on decision consensus.

**Author Contributions:** The individual contribution and responsibilities of the authors are as follows: G.C. and L.W. together designed the research and carried out the study, J.F. helped to draft and revise the manuscript, and C.L. and G.Z. supervised the study and provided suggestions for the revision of the manuscript. All authors have read and agreed to the published version of the manuscript.

**Funding:** This research was funded by the National Natural Science Foundation of China (No. 71761006).

**Acknowledgments:** Summa Maidstone is greatly appreciated by all authors for her language editing and polishing works.

**Conflicts of Interest:** The authors declare no conflict of interest.

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
