# Peer review of "A Large Group Emergency Decision-Making Method Based on Uncertain Linguistic Cloud Similarity Method"

_mca, doi:10.3390/mca27060101_

Round 1

Reviewer 1 Report

Summary:

The authors of this paper proposed a trust update technique-based uncertain linguistic
cloud similarity method for emergency decision-making. To begin, they converted the
linguistic preferences into clouds
 in order to exploit the randomness and fuzziness of the received data. Cloud similarity
is used to categorize large-scale decision-makers. Second, decision maker weights are
determined by an improved PageRank algorithm based on trust relationship, while group
weights are determined by a combined weighting method. As a result, large-scale
decision-maker integrated clouds are purchased. Finally, they calculated
the expected relative distance to determine the level of agreement between alternatives.
 To ensure that the consensus level effectively reached the threshold, a consensus-reaching
approach with opinion interaction and trust updating mechanisms was developed.
The consensus level and trust relationship are considered by the opinion interaction mechanism,
 and similarity, consensus level, and cooperation willingness are considered by the
trust updating mechanism. Finally, the total similarity of alternatives was calculated
 in order to determine the best cloud. The proposed method was validated numerically
 to reach a consensus faster and to distinguish the best alternatives more effectively.

Overall, the studied problem is relevant and fits the journal scope,
the paper contains some scientific merit.
However, the paper writing should be improved.
More structural results may be added to strengthen the results.

Comments:

** The abstract should be improved. In fact, 1.    The abstract does not reflect the importance &
findings of the study and need to be strengthened based on the aim/outcomes of the manuscript.
 Authors may consider the following structure: Introduction,
Research Goals, Research Method, Results, Conclusions, Implications, Originality.
Please give numerical results and implications in the abstract.

** It's true that authors reviewed some previous research works from literature, however, I think that
including a dedicated section for "literature review" is necessary. Importantly, this new section should
clearly position the current work with respect to previous research. For this, I invite authors to provide
a table summarizing contributions of previous work and then the gap filled by their paper.
The objective is to provide a comparison with the previous studies to highlight the research gaps
 and contributions. The theoretical framework needs to be strong.
Please clarify the difference between this work and the previous studies in literature.
Please add more reference (year 2021 and 2022)

** More structural results should be added to strengthen the contribution of the paper. E.g. More results
related to sensitivity analysis, rank reversal, ensemble ranking, etc.

** It is necessary to expand the conclusion. Please add the limitations and the perspectives of this study in the conclusion

Reviewer 2 Report

This paper studies a Large Group Emergency Decision-Making Method Based on Uncertain Linguistic Cloud Similarity Method. It is interesting.

The English writing should be improved with help of professionals. There are some typos, grammatical errors and unsmooth expressions.

The literature review is not extensive. On large Group Emergency consensus Decision-Making, some related and important references are missing. The following may be helpful:

Applied Soft Computing, 110 (2021) 107757; Expert Systems with Applications, 191 (2022) 116328; Applied Soft Computing, 107 (2021) 107383; Information Sciences, 582 (2022) 797-832.

It is necessary to make an overall review of literature to grasp the status of research.

In section 3.5, the convergence of consensus process should be proved mathematically.

Please add some solid comparative analyses in section 4.3.

Reviewer 3 Report

This paper proposes a novel dynamic, trust-driven cloud similarity decision-making method based on large-group consensus and uncertain linguistic information to effectively solve emergency decision-making problems. This paper transforms the linguistic preferences into clouds to take advantage of the randomness and fuzziness of the received information. The opinion interaction mechanism takes account of the consensus level and trust relationship, and the trust updating mechanism considers similarity, consensus level and cooperation willingness. A case of emergency plan selection for epidemic prevention and control illustrates the usefulness of the decision-making method proposed in this study. In general, this paper is innovative, but some details need to be modified. The problems of this paper are as follows:

1.      In line 78-94, the introduction mentions the literature summary related to social networks. However, this paper only considered the trust relationship between experts in social networks, and did not make a specific analysis of social networks, nor did it integrate social network analysis into the consensus model. It is suggested to highlight the trust relationship between experts in the social network in the introduction description and add the social network structure diagram between experts in the case.

2.      In line 386-388, it is suggested that the result of the consensus level of all alternatives be tabulated to prove that only alternative x2 has reached the consensus.

3.      It is suggested to check the mathematical notation carefully. For example, in line 314, the lower limit of expert i’s consensus confidence should be uli.

4.      The lines in Figure 2-5 are not very clear. It is suggested to change the color of the lines or bold the lines to make the experimental results clearer and more intuitive

Round 2

Reviewer 1 Report

The authors revised the paper according to my previous comments.

I invite authors to add a paragraph in the conclusion to include perspectives for future works.

Reviewer 2 Report

The authors have revised the manuscript well according to my previous comments. It can be accepted now.